# Characterization of aerosol particles at Cape Verde close to sea and cloud level heights - Part 2: ice nucleating particles in air, cloud and seawater

Xianda Gong[1], Heike Wex[1], Manuela van Pinxteren[1], Nadja Triesch[1], Khanneh Wadinga Fomba[1], Jasmin Lubitz[1], Christian Stolle[2,3], Tiera-Brandy Robinson[3], Thomas Müller[1], Hartmut Herrmann[1], and Frank Stratmann[1]

[1]Leibniz Institute for Tropospheric Research, Leipzig, Germany
[2]Leibniz-Institute for Baltic Sea Research Warnemünde (IOW), Rostock, Germany
[3]Institute for Chemistry and Biology of the Marine Environment, University of Oldenburg, Wilhelmshaven, Germany

**Correspondence:** Xianda Gong (gong@tropos.de)

**Abstract.** Ice nucleating particles (INPs) in the troposphere can form ice in clouds via heterogeneous ice nucleation. Yet, atmospheric number concentrations of INPs ($N_{\mathrm{INP}}$) are not well characterized and although there is some understanding of their sources, it is still unclear to what extend different sources contribute, nor if all sources are known. In this work, we examined properties of INPs at Cape Verde from different environmental compartments: namely, the oceanic sea surface microlayer (SML), underlying water (ULW), cloud water and the atmosphere close to both sea and cloud level.

Both enrichment and depletion of $N_{\mathrm{INP}}$ in SML compared to ULW were observed. The enrichment factor (EF) varied from roughly 0.4 to 11, and there was no clear trend in EF with ice nucleation temperature.

$N_{\mathrm{INP}}$ in $PM_{10}$ sampled at Cape Verde Atmospheric Observatory (CVAO) at any particular ice nucleation temperature spanned around 1 order of magnitude below $-15$ °C, and about 2 orders of magnitude at warmer temperatures ($>-12$ °C). Among the 17 $PM_{10}$ samples at CVAO, three $PM_{10}$ filters showed elevated $N_{\mathrm{INP}}$ at warm temperatures, e.g., above 0.01 $L^{-1}$ at $-10$ °C. After heating samples at 95 °C for 1 hour, the elevated $N_{\mathrm{INP}}$ at the warm temperatures disappeared, indicating that these highly ice active INPs were most likely biological particles.

$N_{\mathrm{INP}}$ in $PM_1$ were generally lower than those in $PM_{10}$ at CVAO. About 83±22%, 67±18% and 77±14% (median±standard deviation) of INPs had a diameter >1 $\mu$m at ice nucleation temperatures of $-12$, $-15$, and $-18$ °C, respectively. $PM_1$ at CVAO did not show such elevated $N_{\mathrm{INP}}$ at warm temperatures. Consequently, the difference in $N_{\mathrm{INP}}$ between $PM_1$ and $PM_{10}$ at CVAO suggests that biological ice active particles were present in the super-micron size range.

$N_{\mathrm{INP}}$ in $PM_{10}$ at CVAO was found to be similar to that on Monte Verde (MV, at 744 m a.s.l) during non-cloud events. During cloud events, most INPs on MV were activated to cloud droplets. When highly ice active particles were present in $PM_{10}$ filters at CVAO, they were not observed in $PM_{10}$ filters on MV, but in cloud water samples, instead. This is direct evidence that these INPs which are likely biological are activated to cloud droplets during cloud events.

For the observed air masses, atmospheric $N_{\mathrm{INP}}$ in air fit well to the concentrations observed in cloud water. When comparing concentrations of both sea salt and INPs in both seawater and $PM_{10}$ filters, it can be concluded that sea spray aerosol (SSA)

only contributed a minor fraction to the atmospheric $N_{INP}$. This latter conclusion still holds when accounting for an enrichment of organic carbon in super-micron particles during sea spray generation as reported in literature.

## 1 Introduction

Ice particle formation in tropospheric clouds can affect cloud properties such as cloud lifetime, their radiative effects on the atmosphere, and the formation of precipitation (Hoose and Möhler, 2012; Murray et al., 2012). Ice crystals in the atmosphere can be formed either via homogeneous nucleation below $-38$ °C or via heterogeneous nucleation aided by aerosol particles known as ice nucleating particles (INPs) at any temperature below 0 °C. Immersion freezing refers to the process when an INP becomes immersed in an aqueous solution e.g., through the process of cloud droplet activation (Vali et al., 2015). Immersion freezing is suggested to be the most important freezing process for mixed phase clouds (Ansmann et al., 2008; Westbrook and Illingworth, 2013), and is the process we will focus on in this study.

Submicron dust particles are recognized as effective INPs below $-20$ °C (Augustin-Bauditz et al., 2014) and super-micron dust particles were reported to be ice active even up to $-10$ °C (Hoose and Möhler, 2012; Murray et al., 2012). Laboratory studies on natural mineral dusts from different regions have been conducted to quantify the particle's ability to nucleate ice (Niemand et al., 2012; DeMott et al., 2015). Mineral dust particles from deserts are composed of a variety of minerals, and K-feldspar is supposed to be more active for ice nucleation than other minerals in the mixed-phase cloud temperature regime (Atkinson et al., 2013; Augustin-Bauditz et al., 2014; Niedermeier et al., 2015). Boose et al. (2016) found that ice activity of desert dust particles at temperatures between $-35$ and $-28$ °C can be attributed to the sum of the feldspar and quartz content. A high clay content, in contrast, was associated with lower ice nucleation activity. In contrast to field measurements, in laboratory studies often separate types of mineral dusts are examined. Different parameterizations have been employed to summarize the mineral dust particle's ice nucleating ability (Niemand et al., 2012; Ullrich et al., 2017).

A few field measurements have been carried out to quantify the ice nucleation properties of desert dust. Based on airborne measurements, DeMott et al. (2003) found that ice nucleating aerosol particles in air masses over Florida had sources from the North African desert. Chou et al. (2011) observed a good correlation between the number concentration of larger particles and INP number concentration ($N_{INP}$) during a Saharan dust event at the Jungfraujoch in the Swiss Alps. Collecting airborne dust over the Saharan desert, Price et al. (2018) observed two orders of magnitude variability in $N_{INP}$ at any particular temperature from $\sim -13$ to $\sim -25$ °C, which was related to the variability in atmospheric dust loading. This desert dust's ice nucleating activity was only weakly dependent on differences in desert sources, i.e., on the differences in mineral composition that particles emitted from different locations in the desert may have. Schrod et al. (2017) found that mineral dust or a constituent related to dust was a major contributor to $N_{INP}$ of the aerosol on Cyprus, and $N_{INP}$ in elevated dust plumes was on average a factor of 10 higher than $N_{INP}$ at ground level, where the dust loading was lower.

Ocean water can be a potential source of INPs (Brier and Kline, 1959). The source of INPs in ocean water might be associated with phytoplankton blooms (Schnell and Vali, 1976). Recently, Wilson et al. (2015) and Irish et al. (2017) found that organic material, with a diameter <0.2 $\mu$m, is the major ice nucleator in the sea surface microlayer (SML). Based on a long-term

measurement of INPs in the marine boundary layer in the south of and around Australia, Bigg (1973) suggested that INPs in ambient air were from a distant land source, or from a stratospheric source, or brought to sea level by convective mixing and possible ocean sources. Schnell and Vali (1976) also suggested a marine source could explain the observations of Bigg (1973). DeMott et al. (2016) found that the ice nucleation activity from laboratory generated sea spray aerosol (SSA) aligned well with

measurements from diverse regions over the oceans. Furthermore, a connection between marine biological activity and $N_{INP}$ was uncovered in their laboratory study (DeMott et al., 2016). In pristine marine conditions, such as the Southern Ocean, SSA was the main source of the INP population, but $N_{INP}$ was relatively low in the Southern Ocean as well as in the clean marine Northeast Atlantic (McCluskey et al., 2018a, b). These field measurements are consistent with the model work by Burrows et al. (2013), which emphasizes the importance of SSA contribution to INPs in remote marine regions.

It is currently still uncertain whether the coarse mode particles or smaller particles are the major source of atmospheric INPs. Vali (1966) found that the diameters of INPs were mostly between 0.1 and 1 $\mu$m. On the high alpine research station Jungfraujoch, Mertes et al. (2007) found that ice residuals were as small as 300 nm and they were mostly present in the submicron particle size range. Simultaneous measurements of $N_{INP}$ and particle number size distributions were used to develop parameterizations in which $N_{INP}$ depends on a temperature dependent fraction of all particles with sizes above 500 nm (DeMott

et al., 2010, 2015). Conen et al. (2017) found INPs at $-8\,°C$ were equally distributed amongst the particles with sizes up to 2.5 $\mu$m and with sizes between 2.5 and 10 $\mu$m. Other field measurements reported that coarse mode particles were more efficient INP, e.g., INPs (mainly bacterial aggregates and fungal spores) occurred in the size range of 2 - 6 $\mu$m (Huffman et al., 2013). Mason et al. (2016) found for Arctic aerosol that $91\pm9\%$, $79\pm17\%$, and $63\pm21\%$ (mean$\pm1$ standard deviation) of INPs had an aerodynamic diameter of $>1$ $\mu$m at ice activation temperatures of $-15$, $-20$, and $-25\,°C$, respectively. Creamean et al.

(2018) also found that super-micron or coarse mode particles are the most proficient INPs at warmer temperatures in the Arctic boundary layer and they might be biological INPs. Concerning biological INP, it should be mentioned that it is well understood by now that these contain macromolecules of only some ten nanometers in size at the most (Pummer et al., 2015). Some of them are easily separated from their carrier (e.g., from pollen and fungal spores, see e.g., Augustin et al., 2013; O'Sullivan et al., 2016, respectively), while others are embedded in the cell membrane (e.g., for bacteria, Hartmann et al., 2013), but based

on the fact that most atmospheric INPs seem to be super-micron in size, as observed in the above cited literature, it seems that most of the biological ice active macromolecules still occur together with their original carrier in the atmosphere.

   Direct measurement of $N_{INP}$ in the cloud water can be used to estimate concentrations of INPs in the air assuming that most INPs activate as CCN. Joly et al. (2014) measured total and biological (i.e., heat-sensitive) INPs between $-5$ to $-14\,°C$ in cloud samples from the summit of Puy de Dôme (1465 m a.s.l., France). Petters and Wright (2015) summarized many INP

spectra obtained from rain water, melted sleet, snow and hail samples at different sampling locations and reported a range of $N_{INP}$ for these precipitation samples. Based on a shipborne measurement of the east coast of Nova Scotia, Canada, Schnell (1977) directly compared $N_{INP}$ in the seawater to that in the fog water and found that $N_{INP}$ in fog water and seawater appeared to vary quite independently of each other. As one part of the here presented study, these field measurement values will be compared with values obtained from our measurement campaign in the framework of the MarParCloud (Marine biological

production, organic aerosol particles and marine clouds: a Process Chain) project.

During the MarParCloud project, samples collected for INPs analysis include: SML and underlying water (ULW) from the ocean upwind of the island; quartz fiber filter samples of atmospheric aerosol, collected on a tower installed at the island shore (inlet height: 42 m a.s.l) and on a mountaintop (inlet height: 746 m a.s.l); and cloud water collected during cloud events on the mountaintop. In this study, we will first discuss $N_{\mathrm{INP}}$ in the SML and ULW. We will then discuss $N_{\mathrm{INP}}$ in the air,

including a comparison of $N_{\mathrm{INP}}$ in $PM_{10}$ and $PM_1$ and a comparison of $N_{\mathrm{INP}}$ close to both sea and cloud level. Lastly, $N_{\mathrm{INP}}$ in the cloud water will be discussed. In addition, we will provide a feasible way to link $N_{\mathrm{INP}}$ in ambient air, ocean water and cloud water. This connection can be drawn only during times when there were cloud events on the mountaintop, together with data on number concentrations on cloud condensation nuclei ($N_{\mathrm{CCN}}$). Respective information was derived and discussed in an accompanying paper (Gong et al., 2019b). For more information about the campaign itself, we refer to an upcoming overview

paper by van Pinxteren et al. (2019).

## 2 Experiment and methods

### 2.1 Sampling sites and sample types

#### 2.1.1 Sampling site

The measurement campaign was carried out on São Vicente island at Cape Verde from 13 September to 13 October, 2017. We

set up three measurement stations at Cape Verde, at the Cape Verde Atmospheric Observatory (CVAO), on Monte Verde (MV) and an Ocean Station (OS). CVAO (16°51′49 N, 24°52′02 W) is located in the northeastern shore of the island of São Vicente, 70 m from the coastline about 10 m a.s.l. Filter samplers were installed on top of a 32 m tower. MV (16°52′11 N, 24°56′02 W) is located on a mountaintop (744 m a.s.l), ∼7 km away to the west of CVAO. Filter samplers were situated on the ground with the inlet 2 m above the bottom, upwind of any installations on the mountaintop. The OS covered an area at ∼16°53′30

N, ∼24°54′00 W, with a distance of at least 5 km from the island. Details on the measurement site and the meteorological conditions can be found in the accompanying paper (Gong et al., 2019b). In short, the conditions at Cape Verde were quite stable, with temperature of on average 26.6 °C at CVAO and 21.2 °C at MV and wind speeds between 0.6 and 9.7 m s$^{-1}$ with directions from the northeast.

In the following, the different samples collected during the campaign are described in detail. All of these samples were

stored at −20 °C right after sampling. After the campaign the long-term storage and transport of the collected samples from Cape Verde to the Leibniz Institute for Tropospheric Research (TROPOS), Germany was carried out in a cooled container at −20 °C. At TROPOS, all samples were again stored frozen at −20 °C until analysis was done. Measurement sites, locations, sample types and additional information are summarized in Tab. 1.

Following the description of the sampling, we will briefly introduce the measurement methods related to INPs, including

freezing devices, $N_{\mathrm{INP}}$ calculation and measurement uncertainties. Note that all the times presented here are in UTC (corresponding to local time +1). For better comparison, all ambient particle number concentrations in this study are given for standard temperature and pressure (STP, 0 °C and 1013.25 hPa).

**Table 1.** Measurement sites, locations, sample types and measurement instruments.

| Measurement site | Location | Sample type | Instrument |
|---|---|---|---|
| CVAO | 16°51′49 N, 24°52′02 W | PM$_1$ quartz fiber filter | INDA |
| | inlet height: 42 m a.s.l | PM$_{10}$ quartz fiber filter | INDA |
| MV | 16°52′11 N, 24°56′02 W | PM$_{10}$ quartz fiber filter | INDA |
| | inlet height: 746 m a.s.l | Cloud water | LINA, INDA |
| OS | ~16°53′30 N, ~24°54′00 W | SML | LINA, INDA |
| | | ULW | LINA, INDA |

### 2.1.2 Seawater sampling

Seawater samples were taken at the OS by using a fishing boat at a distance of at least 5 km from the coast (off-shore samples). The SML samples were collected using a glass plate sampler (Harvey and Burzell, 1972; Irish et al., 2017; van Pinxteren et al., 2017). The glass plate had a surface area of 2000 cm$^{-2}$ and was immersed vertically into the ocean and then withdrawn at a slow rate (between 5 to 10 cm s$^{-1}$) and allowed to drain for less than 5 s. The surface film adhering to the surface of the glass was scraped off from both sides of the glass plate with a framed Teflon wiper into a 1 liter glass bottle. For each SML sample, several liters were collected and 1 liter required ~55 dips. Based on the amount of material collected, the number of dips and the area of the plate, the averaged thickness of the layer collected was calculated as ~91.0 $\mu$m. ULW samples were collected at the same time and location as the SML samples. ULW was collected from a depth of 1 m by a glass bottle mounted on a telescopic rod in order to monitor sampling depth. The bottle was opened underwater at the intended sampling depth with a specifically designed seal-opener. After collection, the glass bottles containing both the SML and ULW samples were kept in a freezer at −20 °C up to the analysis. During the campaign, 9 SML and 9 ULW samples were collected for INP analysis. Details of SML and ULW samples, including the sampling time, location, salinity and additional information are provided in the supplement, Tab. S1.

### 2.1.3 Aerosol particle sampling

Particle sampling was done using high-volume samplers with either a PM$_{10}$-inlet and or a PM$_1$-inlet (Digitel filter sampler DHA-80, Walter Riemer Messtechnik, Germany) operating with an average flow rate of ~500 L min$^{-1}$ for 24 hours sampling periods. The high-volume samples were collected on 150 mm in diameter quartz fiber filters (Munktell, MK 360) with an effective sampling area of 140 mm in diameter. The filters were preheated in our laboratory at 110 °C for 24 hours to remove the organic carbon background. After sampling, the filters were transported to a freezer where they were kept at −20 °C. For INP analysis, a circular piece of these filters of 2 cm in diameter was used from which then smaller pieces were punched out for the analysis (see Sect. 2.2). From CVAO, there were 17 and 19 filters from PM$_{10}$ and PM$_1$ collection (CVAO PM$_{10}$ and

CVAO PM$_1$), respectively, and at MV, 17 filters were collected for PM$_{10}$ (MV PM$_{10}$). Field blind filters were obtained by inserting clean filters into the Digitel sampler for a period of 24 hours without loading them. Three blind filters were collected during this campaign. Details of filter samples, including sampling time, duration, total volume and additional information can be found in the supplement, Tab. S2 (CVAO PM$_{10}$), Tab. S3 (CVAO PM$_1$) and Tab. S4 (MV PM$_{10}$).

## 2.1.4 Cloud water sampling

During the campaign, MV was in clouds roughly 58% of the time (a detailed analysis on this can be found in Gong et al. (2019b)). Cloud water was collected with CASCC2 (Caltech Active Strand Cloud Collector Version 2) at MV. All cloud drop sizes were collected in one bulk sample. Drops were collected by inertial impaction on Teflon strands with a diameter of 508 $\mu$m. The 50% lower size cut for the CASCC2 was approximately 3.5 $\mu$m diameter. The flow rate through the CASCC2 was approximately 5.8 m$^3$ min$^{-1}$. The CASCC2 is described in more details in Demoz et al. (1996). Between cloud events, the cloud water sampler was cleaned with a large amount ($\sim$5 L) of ultrapure water. Once the collector was cleaned, a blank was taken by spraying about 200 mL of ultrapure water into the collection strands in the collector and subsequent sampling of this water. After collection, the cloud water samples were kept in a freezer at $-20$ °C. During the campaign, 13 cloud samples were collected for INP analysis. The details of cloud samples, including sampling time, duration, volume and additional information are provided in the supplement, Tab. S5.

## 2.2 Freezing devices

Two droplet freezing devices called LINA (Leipzig Ice Nucleation Array) and INDA (Ice Nucleation Droplet Array) have been set up at TROPOS in Germany. The design of LINA was inspired by Budke and Koop (2015). Briefly, 90 droplets with the volume of 1 $\mu$L were pipetted from the samples onto a thin hydrophobic glass slide, with each droplet being placed separately into its own compartment. After pipetting, the compartments were sealed at the top with another glass slide, to prevent the droplets from evaporation and to prevent ice seeding from neighboring droplets. The droplets were cooled on a Peltier element with a cooling rate of 1 K min$^{-1}$ down to $-35$ °C, while the setup was illuminated by a circular light source from above. Once the cooling started, pictures were taken every 6 s by a camera. The number of frozen versus unfrozen droplets was derived automatically by an image identification program in Python. LINA was employed to measure SML, ULW and cloud water samples in this study. More detailed parameters and the temperature calibration of LINA and its application can be found in previous studies (Chen et al., 2018; Gong et al., 2019a).

The design of INDA was inspired by Conen et al. (2012), but deploying PCR-trays instead of separate tubes. For quartz fiber filters, circular pieces with a diameter of 1 mm were punched out. Each of the 96 wells of a PCR-tray were filled with the filter piece together with 50 $\mu$L of ultrapure water. For SML, ULW and cloud water samples, 50 $\mu$L of the water samples was filled into each PCR-tray. After sealing by a transparent foil, the PCR-tray was placed on a sample holder and immersed into a bath thermostat, where it was illuminated from below with a LED light source. The bath thermostat then decreased the temperature with a cooling rate of approximately 1 K min$^{-1}$. Real-time images of the PCR-tray were recorded every 6 s by a CCD (charge-coupled device) camera. Frozen droplets can be identified based on the brightness change during the freezing

process. A program recorded the actual temperature of the cooling bath and related it to the real-time images from the CCD camera. The temperature in the PCR-trays had been calibrated. More detailed parameters and temperature calibration of INDA and its application can be found in previous studies (Chen et al., 2018; Hartmann et al., 2019).

## 2.3 Deriving $N_{\text{INP}}$

### 2.3.1 Basic calculation

Based on Vali (1971), the cumulative concentration of INP ($N_{\text{INP}}$) as a function of temperature per air or water volume can be calculated by:

$$N_{\text{INP}}(\theta) = \frac{-\ln(1 - f_{\text{ice}}(\theta))}{V} \tag{1}$$

with

$$f_{\text{ice}}(\theta) = \frac{N(\theta)}{N_{\text{total}}} \tag{2}$$

where $N_{\text{total}}$ is the number of droplets and $N(\theta)$ is the number of frozen droplets at temperature $\theta$. Equation 1 accounts for the possibility of the presence of multiple INPs in one vial by assuming that INPs are Poisson distributed. This way, the cumulative number of INP active at any temperature will be obtained although only the most ice active INP (nucleating ice at the highest temperature) present in each droplet/well will be observed. As for the quartz fiber filters, V is the volume of air collected onto one circular 1 mm filter piece placed in each well, resulting in airborne $N_{\text{INP}}$. The information of the air volume can be found in the supplement, Tab. S2, Tab. S3 and Tab. S4. As for the SML, ULW and cloud water, V is the volume of droplet/well ($V_{\text{LINA}}$=1 $\mu$L, $V_{\text{INDA}}$=50 $\mu$L), resulting in $N_{\text{INP}}$ per volume of water. Compared to the droplets examined in a LINA measurement, INDA measurements have a larger volume of water in each well. The larger volume of water corresponds to a higher probability of the presence of INPs in each well, therefore INDA can detect INPs at warmer temperatures, where INP are more scarce. In this study, the derived $N_{\text{INP}}$ from LINA and INDA measurements were combined when both instruments were deployed.

### 2.3.2 Uncertainty and background

Because the number of INPs present in the water is usually small (some single up to a few tens of INPs per examined droplet/well), and the number of droplets/wells considered in our measurements is limited, statistical errors need to be considered in the data evaluation. Therefore, confidence intervals for $f_{\text{ice}}$ were determined using the method suggested by Agresti and Coull (1998). These confidence intervals were estimated according to the improved Wald interval which implicitly assumes a normal approximation for binomially distributed measurement errors. Previous studies (McCluskey et al., 2018a; Suski et al., 2018; Gong et al., 2019a) used the same method to calculate the freezing devices' measurement uncertainties.

For the quartz fiber filters, a background freezing signal resulting from the field blind filters was determined by doing a regular INDA measurement with these filters. Measured $N_{\text{INP}}$ from the sampled filters was corrected by subtracting the averaged background concentrations determined for the blind filters, as explained in Wex et al. (2019). All values for airborne

$N_{\text{INP}}$ presented in the following are background-corrected. A detailed description of the background subtraction method and background values are provided in the supplement. For those samples that were already collected in a liquid state (ULW, SML and cloud water ), a background correction was not done.

### 2.3.3 Salinity correction of SML and ULW

SML and ULW samples were adjusted to account for the freezing depression caused by dissolved salts in sea water. Based on Kreidenweis et al. (2005), the water activity can be calculated by:

$$a_{\text{w}} = \frac{n_{\text{water}}}{n_{\text{water}} + i * n_{\text{solute}}} \tag{3}$$

where the $n_{\text{solute}}$ and $n_{\text{water}}$ are the number of moles of solute and water in solution, respectively. $i$ is the van't Hoff factor (Pruppacher and Klett, 2010). We assumed sea salt to be mainly sodium chloride, for which the van't Hoff factor is 2. The freezing depression temperature as a function of $a_{\text{w}}$ was taken from Koop and Zobrist (2009). In our study, this was roughly a correction by 2.2 °C.

### 2.4 Active surface site density

A thorough analysis of particle number size distributions (PNSDs) has been presented in Gong et al. (2019b), and based on these PNSDs we derived the particle surface area size distributions (PASDs) for use in this study (to be seen in the supplement, Fig. S14). These PASDs were used to determine the temperature-dependent cumulative active surface site density ($n_{\text{s}}$) for aerosol particles. The $n_{\text{s}}$ is a measure of how well an aerosol acts as a seed surface for ice nucleation. The $n_{\text{s}}$ can be calculated as:

$$n_{\text{s}} = \frac{N_{\text{INP}}(\theta)}{A_{\text{total}}} \tag{4}$$

where $A_{\text{total}}$ is the concentration of the total particle surface area.

For cases where a single type of aerosol, such as one type of mineral dust, is examined in laboratory studies, $A_{\text{total}}$ can be the total particle surface area. However, when field experiments are done, using the total particle surface area of the atmospheric aerosol assumes that all particles contribute to INP and have the same $n_{\text{s}}$, while the vast majority of these particles will not even be an INP. On the other hand, singling out the contribution of separate INP types in the atmospheric aerosol and relying $n_{\text{s}}$ only to them by using their contribution to the total surface area is at least demanding if not often impossible. This has to be kept in mind when interpreting heterogeneous ice nucleation in terms of $n_{\text{s}}$. An example of separating the $n_{\text{s}}$ for dust and marine ambient air can be found in Cornwell et al. (2019).

## 3 Results

### 3.1 INP in SML and ULW

Based on Eq. 1, the derived $N_{\text{INP}}$ in seawater as a function of temperature is shown in Fig. 1, for both SML and ULW. Note that for each sample a separate INP spectrum is shown. Error bars show the 95% confidence interval. For completeness, $f_{\text{ice}}$ of all seawater samples is shown in the supplement, Fig. S1 (measured by LINA) and Fig. S2 (measured by INDA). The variation of $N_{\text{INP}}$ at any particular temperature is within one order of magnitude. Included in Fig. 1 are previous studies of $N_{\text{INP}}$ measured east of Greenland in the Arctic (shown as red box) and east of America in the North Atlantic Ocean (shown as black box) from Wilson et al. (2015).

The concentration range detected for ULW in Wilson et al. (2015) (both in the Arctic and the North Atlantic Ocean) roughly agrees with our data. In Wilson et al. (2015), $N_{\text{INP}}$ in the SML in the North Atlantic Ocean is at the lower end of that found in the Arctic. A possible reason for this difference could be the biological activity of the ocean water. Wilson et al. (2015) found that organic material was correlated to $N_{\text{INP}}$ in SML, and that $N_{\text{INP}}$ per gram of total organic carbon in the Arctic and the North Atlantic Ocean were comparable. A recent study found that the SML at Cape Verde was oligotrophic, which is supported by the low Chlorophyll-a and transparent exopolymer particles concentrations found during the MarParCloud campaign (Robinson et al., 2019). The low biological activity in the SML around Cape Verde could be the reason why $N_{\text{INP}}$ in SML in this study is lower than those reported in Wilson et al. (2015).

To better quantify the enrichment or depletion of $N_{\text{INP}}$ in SML to ULW, we derived an enrichment factor (EF). An enrichment might be expected as organic material is known to attach to air bubbles rising to the ocean surface. The EF in SML was calculated by dividing $N_{\text{INP}}$ in SML ($N_{\text{INP, SML}}$) by the respective $N_{\text{INP}}$ measured in ULW ($N_{\text{INP, ULW}}$), as the below equation shows:

$$\text{EF} = \frac{N_{\text{INP, SML}}}{N_{\text{INP, ULW}}} \tag{5}$$

Enrichment of $N_{\text{INP}}$ in the SML is indicated when EF > 1, while depletion is indicated when EF < 1. Figure 2 shows the EF as a function of the temperature at which $N_{\text{INP}}$ was determined in the freezing devices. Both enrichment and depletion were observed, but there is no clear trend of the EF with temperature. Most of the variation seen here is likely caused by measurement uncertainties, which are indicated in Fig. S3 in the supplement. EF varied from 0.36 to 11.40 at $-15\,^{\circ}\text{C}$ and from 0.36 to 7.11 at $-20\,^{\circ}\text{C}$. By comparing $T_{10}$ (the temperature at which 10% of droplets had frozen) for the SML and ULW, Wilson et al. (2015) observed higher enrichment of INPs in SML in both the Arctic and the North Atlantic Ocean. However, Irish et al. (2017) observed both enrichment and depletion of INPs in SML in the Arctic, similar to the observation made in the present study.

These differences in EF between studies might partially be due to differences in the techniques deployed and different SML thickness in our and the other studies. SML samples were estimated to be about ~91.0 μm thick in this study, while for Wilson et al. (2015) those were between 6 to 83 μm. It is interesting to note that we used glass dipping for the samples analyzed herein, while both glass dipping and a rotating drum sampler were used in Wilson et al. (2015). Previous studies pointed out that

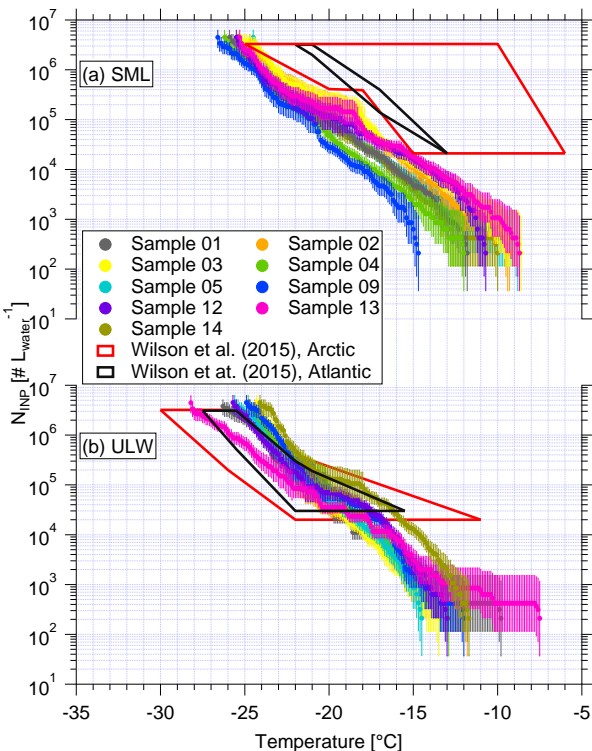

**Figure 1.** $N_{\text{INP}}$ as a function of temperature in SML (a) and ULW (b). Error bars show the 95% confidence interval. Previous field measurements of $N_{\text{INP}}$ in seawater by Wilson et al. (2015) are compared, as shown by red and black boxes.

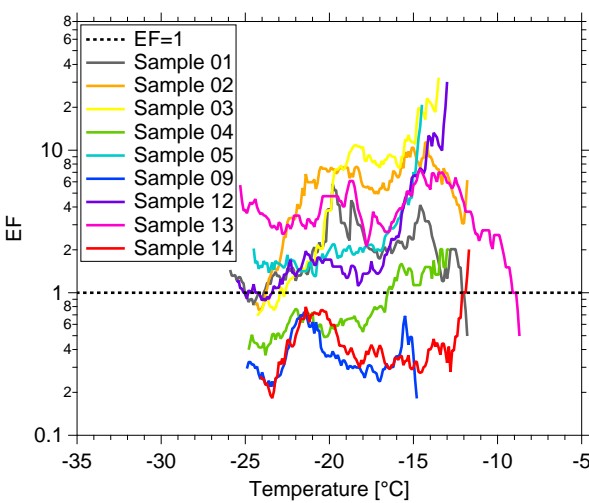

**Figure 2.** Enrichment factor (EF) as function of ice nucleation temperature. The EF=1 is shown by dashed line.

rotating drum sampler and the glass dipping method probe different thicknesses of the SML, thus making a direct comparison of both SML thickness as well as enrichment factors generally difficult (Agogué et al., 2004; Aller et al., 2017).

## 3.2 $N_{\text{INP}}$ in air

Three different sets of filter samples were collected at CVAO and MV, i.e., CVAO PM$_{10}$, CVAO PM$_1$ and MV PM$_{10}$. In this section, we will discuss $N_{\text{INP}}$ at CVAO for the two different size classes and compare $N_{\text{INP}}$ from close to the sea level (CVAO) to that at cloud level (MV).

### 3.2.1 $N_{\text{INP}}$ close to sea level

**CVAO PM$_{10}$**

$N_{\text{INP}}$ as a function of temperature from CVAO PM$_{10}$ filters and CVAO PM$_1$ filters are shown in Fig. 3(a) and (b). Error bars show the 95% confidence interval. The respective values of $f_{\text{ice}}$ are shown in the supplement, Fig. S4 (CVAO PM$_{10}$) and Fig. S8 (CVAO PM$_1$), together with the results from the blind filters. The CVAO PM$_{10}$ filter samples were all active at $-11.3$ °C and the highest freezing temperature was found to be $-5.0$ °C. Filter samples collected in Cape Verde over the period 2009-2013 for INP measurement were reported by Welti et al. (2018), and they are shown as gray background in Fig. 3(a). The measured $N_{\text{INP}}$ in this study is within the $N_{\text{INP}}$ range presented by Welti et al. (2018).

$N_{\text{INP}}$ at any particular temperature span around 1 order of magnitude below $-15$ °C, and about 2 orders of magnitude at warmer temperatures. This is consistent with the previous studies from O'Sullivan et al. (2018) and Gong et al. (2019a), who carried out field measurement in northwestern Europe and the eastern Mediterranean, respectively. A few samples (CVAO 1596, CVAO 1641 and CVAO 1643) showed elevated concentrations above 0.01 L$^{-1}$ at $-10$ °C. Biological particles usually contribute to INPs at this moderate supercooling temperature (Kanji et al., 2017; O'Sullivan et al., 2018).

Biological INPs contain specific ice-nucleating proteins. These proteins are disrupted and denatured by heating which causes them to lose their ice-nucleating ability. However, the inorganic ice-nucleating material, such as dust particles, is insensitive to heat (Wilson et al., 2015; O'Sullivan et al., 2018). Therefore, a commonly used heat treatment was deployed to assess the contribution of biological INPs to the total INPs in this study. Samples CVAO 1596, CVAO 1641 and CVAO 1643 were heated to 95 °C for 1 hour and the resulting $N_{\text{INP}}$ are shown in Fig. S6. A clear comparison of before and after heating $f_{\text{ice}}$ is shown in Fig. S7. A large reduction of more than one order of magnitude in $N_{\text{INP}}$ at T>$-15$ °C was observed in the samples after heating. The reductions in $N_{\text{INP}}$ became smaller at colder temperature and were, for example, less than one order of magnitude at T=$-20$ °C. This shows that biological aerosol contributed a large fraction of total INPs in PM$_{10}$ at T>$-20$ °C.

The correlation of $N_{\text{INP}}$ at different temperatures within one sample was calculated, by comparing each $N_{\text{INP}}$ at each temperature to that at each other temperature at which a measurement had been made. That was done separately for each of the samples. For temperature steps of 0.1 °C, $N_{\text{INP}}$ at every temperature was correlated to that at every other temperature in the measurement range. With increasing difference in temperatures, the variation in $N_{\text{INP}}$ at two temperatures become less correlated. As long as the examined temperature difference was less than 2 °C, $N_{\text{INP}}$ were correlated. But when looking at this

in a broader picture, in the temperature region down to $\sim -16.8\ °C$, $N_{INP}$ at all temperatures correlated well with that at all other temperatures, with coefficient of determination ($R^2$) > 0.8 and p < 0.01. The same was true for $N_{INP}$ in the temperatures region < $-18.4\ °C$. In between these two temperature regimes (between > $-16.8\ °C$ and < $-18.4\ °C$), the correlation of $N_{INP}$ was clearly lower. Therefore, it might be expected that INPs that are active in these two temperature regimes originated from different sources.

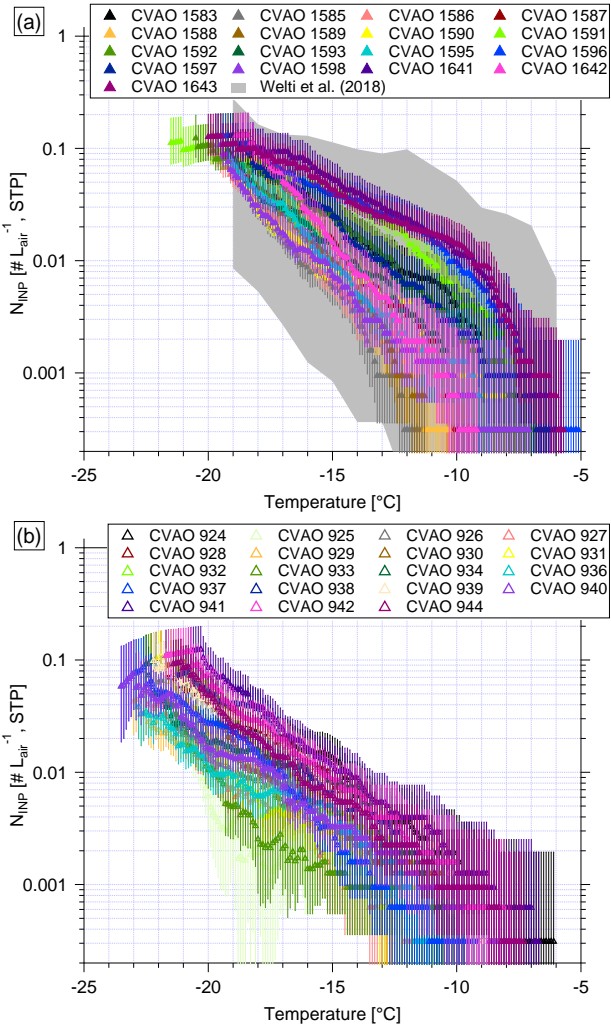

**Figure 3.** $N_{INP}$ as a function of temperature from CVAO PM$_{10}$ filters (a) and CVAO PM$_1$ filters (b). The field measurement of $N_{INP}$ in PM$_{10}$ by Welti et al. (2018) is shown by gray shadow in Fig. (a). Error bars show the 95% confidence interval.

**CVAO PM$_1$ in comparison to CVAO PM$_{10}$**

$N_{\text{INP}}$ in PM$_1$ filters are also determined in this study (as shown in Fig. 3(b)). An initial inspection of the data shows that the bulk of the data of $N_{\text{INP}}$ for CVAO PM$_1$ is below that for CVAO PM$_{10}$. Comparing $N_{\text{INP}}$ for PM$_1$ and PM$_{10}$, two key features are evident :

5    1. Larger particles, i.e., super-micron ones, were more efficient INPs, which is independent of temperature in the examined range.

    2. Smaller particles, i.e., submicron ones, exhibited an equal spread of about 1 order of magnitude in $N_{\text{INP}}$ for the whole temperature range (see Fig. 3(b)). The elevated $N_{\text{INP}}$ at warm temperatures which are seen for CVAO PM$_{10}$ are not observed for CVAO PM$_1$.

10    As for the first feature, we calculated the ratio of $N_{\text{INP}}$ in super-micron size range to $N_{\text{INP}}$ in PM$_{10}$ during the same time period and found that $83\pm22\%$, $67\pm18\%$ and $77\pm14\%$ (median$\pm$standard deviation) of INPs had a diameter of $>1$ $\mu$m at ice activation temperatures of $-12$, $-15$, and $-18\,^\circ$C, respectively. On average, over all temperatures, this INP number fraction for super-micron particles is roughly 70% (shown for a higher temperature resolution in Fig. 4), almost independent of temperature. Mason et al. (2016) and Creamean et al. (2018) also found that the majority of INPs is in the super-micron size range. However,

15    they see even increasing fractions towards higher temperatures. For the present study, as said above, only three of the examined 17 filters showed clearly elevated $N_{\text{INP}}$ at high temperatures, so overall such an increase was not observed.

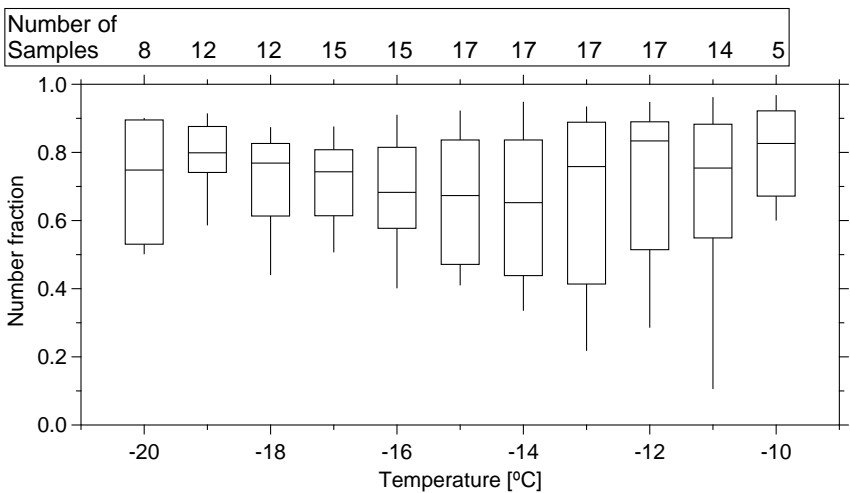

**Figure 4.** Boxplot of number fraction of INPs in the size range of $>1$ $\mu$m as a function of temperature. The boxes represent the interquartile range. Whiskers represent 10th to 90th percentile. The number of samples indicated on top of the figure shows how many different samples contributed at the different temperatures.

As for the second feature, looking at Fig. 3(b), we found that $N_{\text{INP}}$ spread about 1 order of magnitude at any temperature from $-12$ to $-20\,^\circ$C. As outlined above, a few PM$_{10}$ samples showed elevated concentrations at warm temperatures, showing

up as a "bump" in the freezing curves at higher temperatures. This bump at warm temperatures was not observed for the CVAO PM$_1$ filters. $N_{INP}$ of CVAO 932, CVAO 942 and CVAO 944 (sampled at the same time as CVAO 1596, CVAO 1641 and CVAO 1643) are all below 0.001 L$^{-1}$ at $-10°$C. As mentioned above, INP active at comparably high temperatures were found to be biological in origin in this study, and the comparison between PM$_{10}$ and PM$_1$ samples show that there are biological INPs in the CVAO PM$_{10}$ samples that are absent in the CVAO PM$_1$ samples, i.e., that the detected biological INPs are super-micron in size. This suggests that these biological INPs might originate from long-range transport, as marine biological INPs were usually reported to be submicron in size (Wilson et al., 2015; Irish et al., 2017). The contribution of SSA to INPs will be discussed further in Sect. 3.4.

### 3.2.2 $N_{INP}$ at cloud level

In the companion paper (Gong et al., 2019b), we discussed PNSD and CCN number concentration ($N_{CCN}$) at CVAO and MV. We found that particles are mainly well mixed in the marine boundary layer and derived the periods with cloud events, with a time resolution of ∼30 minutes, at MV. In the present study, $N_{INP}$ in PM$_{10}$ at CVAO and MV are compared. The fraction of time during which there was a cloud event to the total sampling time (cloud time fraction) for each filter is summarized in the supplement, Tab. S4. All of the filters were affected by cloud events with a cloud time fraction from 4.17 to 100%, with two filters being affected only little (cloud time fraction <10%), i.e., MV 1602 and MV 1603. When comparing results from these two filters to those from filters sampled at the same time at CVAO (see Fig. 5(a)), we found that $N_{INP}$ are quite similar close to sea level (CVAO) and cloud level (MV). This is in line with what was discussed in the companion paper (Gong et al., 2019b), i.e., the marine boundary is often well mixed at Cape Verde.

Figure 5(b) compares $N_{INP}$ at CVAO and MV when MV filters were mostly collected during cloud events with cloud time fractions >90%. During the cloud events, the filters did not collect droplets larger than 10 $\mu$m because of the inlet cutoff. It is obvious from Fig. 5 that for these cases, $N_{INP}$ at MV is much lower than that at CVAO, implying that particularly INPs that were ice active above $\sim -17$ ° C were activated to cloud droplets to a large degree. But note that even when filters have a cloud time fraction of 100% (MV 1615 and MV 1616), the respective filters still had clearly more INPs on them than the field blind filters (see supplement, Fig S9). This might indicate that either not all INPs are activated to cloud droplets, or, on the other hand, that some INPs were only recently activated to a cloud droplet and the droplet size was smaller than 10 $\mu$m. These observations are consistent with results by Siebert and Shaw (2017) who observed broad cloud droplet size distributions in a size range from ∼5 to 25 $\mu$m in shallow cumulus clouds, with the maximum of the distribution still being below 10 $\mu$m.

Concerning the super-micron particles of likely biological origin that activated ice already at $-10$ ° C and above, it is observed that the related corresponding bump is not seen in the corresponding data from MV (MV 1610, MV 1614 and MV 1616 - to be seen in the supplement, Fig. S10). This indicates that these INPs were all activated to cloud droplets during the cloud events, and we will come back to this below.

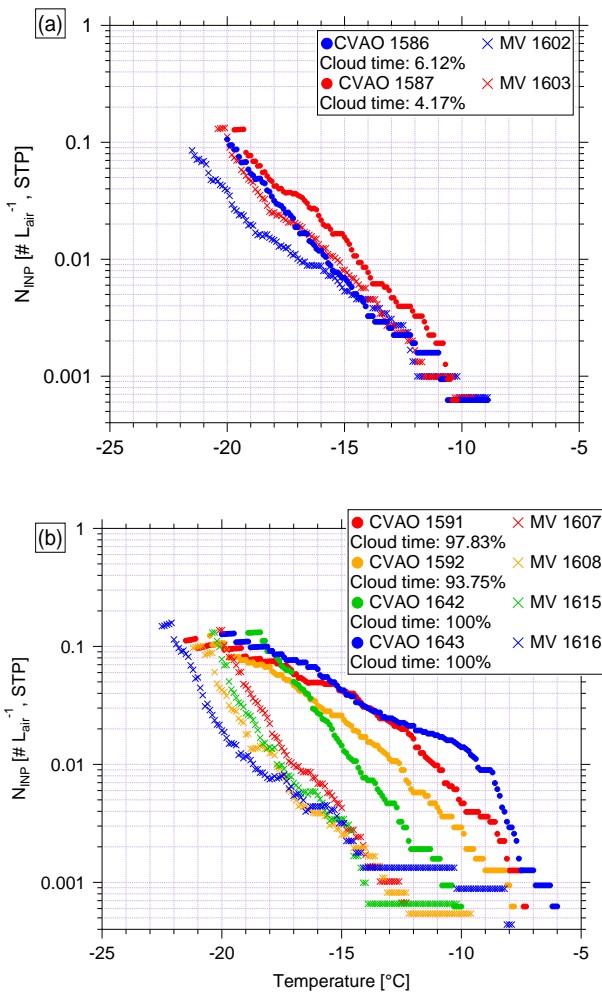

**Figure 5.** $N_{INP}$ as a function of temperature from CVAO $PM_{10}$ filters and MV $PM_{10}$ filters during (a) less (cloud time fraction <10%) cloud effected periods and (b) highly (cloud time fraction >90%) cloud effected periods.

### 3.3 INP in cloud water

#### 3.3.1 Main characteristics and $N_{INP}$ in cloud water

Thirteen cloud water samples were collected during cloud events in this study. Sampling durations varied from 2.5 to 13 hours and volumes varied from 78 to 544 mL. The most abundant inorganic species were $Na^+$ and $Cl^-$, followed by $SO_4^{2-}$, $NO_3^-$
and $Mg^{2+}$. For example, the mass concentration of $Na^+$ and $Cl^-$ varied from 5.00 to 46.11 and 9.27 to 70.30 mg $L^{-1}$, with a mean value of 17.31 and 28.86 mg $L^{-1}$, respectively. Somewhat different values which are still roughly in the same range were reported by Gioda et al. (2009), who found in Puerto Rico the $Na^+$ and $Cl^-$ concentration in the cloud water varied from

3.79 to 15.53 and 5.90 to 23.20 mg L$^{-1}$, with a mean of 10.74 and 15.67 mg L$^{-1}$, respectively. All of the above mentioned parameters are summarized in the supplement, Tab. S5.

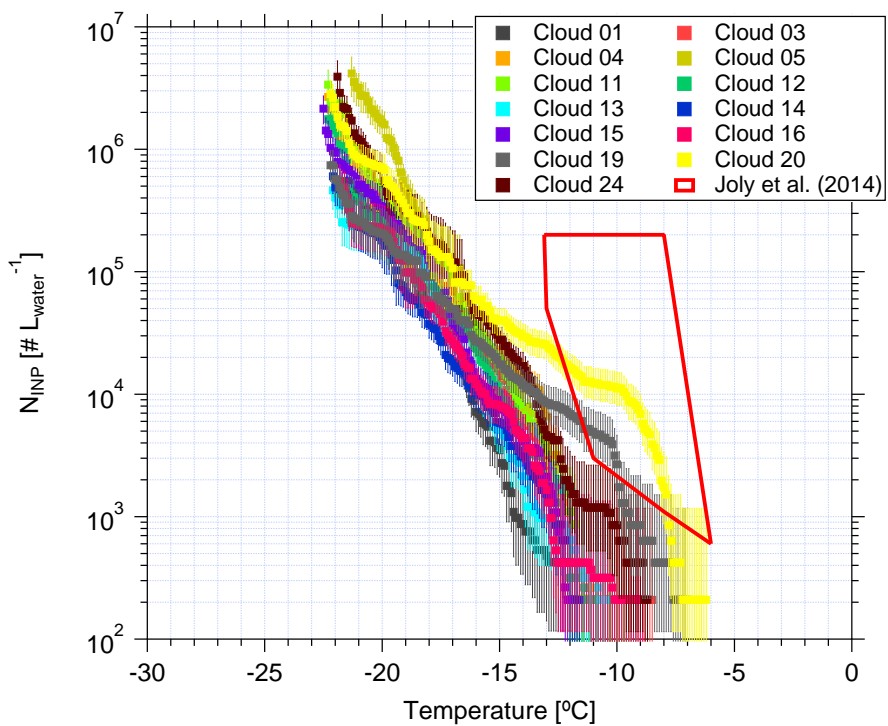

**Figure 6.** $N_{\text{INP}}$ in cloud water as a function of temperature. Error bars show the 95% confidence interval. Previous field measurements of $N_{\text{INP}}$ in cloud water by Joly et al. (2014) are shown as red box for comparison.

Based on Eq. 1, the derived $N_{\text{INP}}$ as a function of temperature is shown in Fig. 6. Error bars represent the 95% confidence interval. For completeness, $f_{\text{ice}}$ for cloud water is shown in the supplement, Fig. S12 (measured by LINA) and Fig. S13 (measured by INDA). $N_{\text{INP}}$ at any particular temperature span less than 1 order of magnitude below $-15\,°$C, while they span 2 orders of magnitude at warmer temperatures. We observed elevated $N_{\text{INP}}$ in the cloud water at warm temperatures (above 1000 L$^{-1}$ at $-10\,°$C) particularly for the Cloud 19, Cloud 20 and Cloud 24 samples. Joly et al. (2014) measured the total and biological (i.e., heat-sensitive) INPs between $-5$ to $-14\,°$C from the summit of Puy de Dôme (1465 m a.s.l., France), as shown in the red box in Fig. 6. Joly et al. (2014) observed very high concentrations of both biological particles and $N_{\text{INP}}$. Agreement of $N_{\text{INP}}$ in cloud water all over the world was not expected, since the sources of INPs are different in different locations.

When highly ice active particles were present for CVAO PM$_{10}$ filters (CVAO 1596, CVAO 1641 and CVAO 1643), they were not observed for MV PM$_{10}$ (MV 1610, MV1614 and MV 1616, which had cloud time fractions of 52, 87 and 100%, respectively), but instead were found in cloud water samples (Cloud 19, Cloud 20 and Cloud 24). This is in line with what was outlined in Sect. 3.2.2 that these highly ice active particles were activated to cloud droplets during cloud events. Periods during which clouds were present at MV, together with the sampling periods of all cloud water samples and selected CVAO

$PM_{10}$ filters (those that had higher $N_{INP}$ at warm temperatures, CVAO 1596, CVAO 1641 and CVAO 1643) can be checked in the supplement, Fig. S11.

### 3.3.2 Connecting INPs in the cloud water with these in the air

In the following, $N_{INP}$ in the cloud water will be compared to that in the air. To be able to do this, we used measured values of $N_{CCN}$ to calculate cloud droplet number concentrations. These, together with an assumption on cloud droplet size ($d_{drop}$) yields the volume of cloud water per volume of air, given as $F_{cloud\_air}$ in Eq. 6:

$$F_{cloud\_air} = N_{CCN} * \pi/6 * d_{drop}^3 \tag{6}$$

For the calculation, we used $N_{CCN}$ measured at CVAO at a supersaturation of 0.30% (Gong et al., 2019b). $N_{CCN}$ was averaged for the different periods when each cloud water sample was collected. The chosen supersaturation corresponds to a critical diameter of roughly 80 nm, which is at the Hoppel minimum of the respective particle number size distributions (Gong et al., 2019b), indicating that this is indeed the relevant supersaturation occurring in the prevailing clouds. Based on previous studies (Miles et al., 2000; Bréon et al., 2002; Igel and Heever, 2017; Siebert and Shaw, 2017), we assumed that $d_{drop}$ varies between 7 and 20 $\mu$m and did separate estimates for these two values and additionally for 15 $\mu$m. The calculation based on this size range of cloud droplets should cover all that can be expected to occur.

Following this approach, $F_{cloud\_air}$ varied from $4.2*10^{-7}$ to $1.1*10^{-6}$, with a median of $8.5*10^{-7}$ $m_{water}^3/m_{air}^3$. To see how reliable these values are, we also examined the following: assuming all sodium chloride particles were activated to cloud droplets, $F_{cloud\_air}$ can be also estimated from the ratio of sodium chloride mass concentration in air to that in cloud water. This ratio varied from $1.1*10^{-7}$ to $4.4*10^{-7}$ $m_{water}^3/m_{air}^3$, which is at the lower end but still comparable to $F_{cloud\_air}$ as we derived it above. Previous studies used the liquid water content (LWC), which is a measure of the mass of the water in a cloud in a specified amount of dry air. Typical ranges for LWC in thicker clouds are between 0.2 and 0.8 g m$^{-3}$ (Rangno and Hobbs, 2005; Petters and Wright, 2015), corresponding to $F_{cloud\_air}$ between $2*10^{-7}$ to $8*10^{-7}$ $m_{water}^3/m_{air}^3$, which again agreed well with the above given values derived for this study.

With this $F_{cloud\_air}$, $N_{INP}$ in the respective volume of air can be compared to $N_{INP}$ in this volume of cloud water when assuming that all INPs are CCN, which, based on the super-micron size of most of the INPs alone, is likely. To do so, $N_{INP}$ obtained for cloud water was multiplied by $F_{cloud\_air}$ (for the three different assumptions on $d_{drop}$) to yield $N_{INP}$ in the air ($N_{INP,air}$), given in Eq. 7:

$$N_{INP,air} = F_{cloud\_air} * N_{INP,cloud} \tag{7}$$

Figure 7 shows the measured $N_{INP}$ in the air as a function of temperature by squares. Derived $N_{INP,air}$ from cloud water (calculated with a $d_{drop}$ of 15$\mu$m) are shown by triangles. The samples with comparatively high numbers of INPs active at warm temperatures, are shown in different colors. CVAO 1596, CVAO 1641 and CVAO 1643 are shown by green squares (the rest shown by blue squares) and derived $N_{INP,air}$ from samples collected for Cloud 19, Cloud 20 and Cloud 24 are shown by brown triangles (the rest shown by red triangles). The range of values indicated for $N_{INP,air}$ was obtained from using 7 and 20 $\mu$m cloud droplet size, with 7 $\mu$m droplets yielding the lower boundary and 20 $\mu$m the upper one.

There is general agreement between measured and derived $N_{\text{INP}}$ in air, however, with some variation where the values derived from cloud water samples are somewhat lower. This might be connected to a less than optimal sampling efficiency of the cloud water sampler, which has a 50% collection efficiency at 3.5 $\mu$m. Also the spread in the derived values, originating from the different assumed $d_{\text{drop}}$, is rather large. Nevertheless, it is striking that at least within an order of magnitude, based on

5 our comparably simple assumptions, an agreement between concentrations of INP in the air and in cloud water is found.

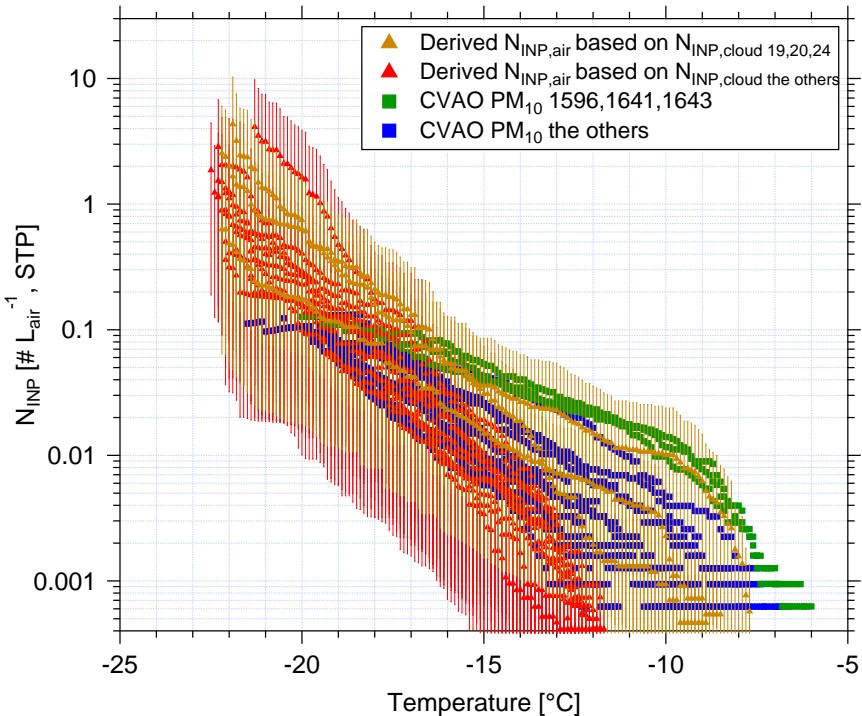

**Figure 7.** The measured atmospheric $N_{\text{INP}}$ as a function of ice nucleation temperature are shown as squares. The derived $N_{\text{INP}}$ in the air ($N_{\text{INP,air}}$) based on INP concentrations measured for cloud water are shown as triangles. The samples with highly ice active INPs at warm temperatures, are shown in a different color than the others: CVAO 1596, CVAO 1641 and CVAO 1643 are shown as green squares and derived $N_{\text{INP,air}}$ based on Cloud 19, Cloud 20 and Cloud 24 are shown as brown triangles. The uncertainty range indicated for the derived $N_{\text{INP,air}}$ originate from calculations with 7 and 20 $\mu$m cloud droplet size.

### 3.4 INPs originating from sea spray

In the following section, it will briefly be discussed whether SSA contributed noticeably to INPs in the air. Assuming sea salt and INPs to be similarly distributed in both, seawater and air (i.e., assuming that INPs would not be enriched during the production of sea spray), $N_{\text{INP}}$ in the air originating from sea spray ($N_{\text{INP}}^{\text{sea spray,air}}$) can be calculated based on Eq. 8:

$$N_{\text{INP}}^{\text{sea spray,air}} = \frac{\text{NaCl}_{\text{mass,air}}}{\text{NaCl}_{\text{mass,seawater}}} * N_{\text{INP}}^{\text{seawater}} \tag{8}$$

where $NaCl_{mass,air}$ and $NaCl_{mass,seawater}$ are sodium chloride mass concentrations in air and seawater, respectively. $N_{INP}^{seawater}$ is the INP number concentration in the seawater (this calculation can be done similarly for both SML and ULW).

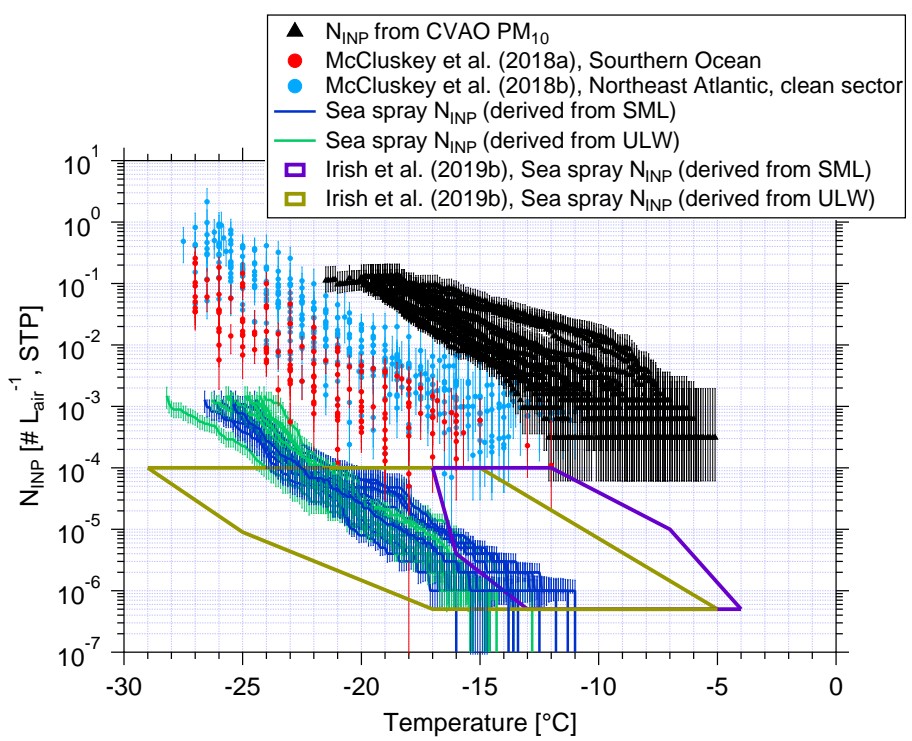

**Figure 8.** Atmospheric $N_{INP}$ are shown as a function of temperature from $PM_{10}$ filters (black triangles), together with error bars showing the 95% confidence interval. $N_{INP}$ as a function of temperature from McCluskey et al. (2018a, b) are shown by red and light blue dots, respectively. Error bars show the 95% confidence interval. $N_{INP}$ in the air originating from sea spray ($N_{INP}^{sea\,spray,air}$) from this study are shown by blue (derived from SML) and green lines (derived from ULW). $N_{INP}^{sea\,spray,air}$ from Irish et al. (2019b) are shown by purple (derived from SML) and brown (derived from ULW) boxes.

$NaCl_{mass,air}$ and $NaCl_{mass,seawater}$ can be found in the supplement, Tab. S1 and Tab. S2. $NaCl_{mass,seawater}$ was very stable, with a median value $\sim$31 g L$^{-1}$. $NaCl_{mass,air}$ showed large variability from 3.40 to 17.76 $\mu$g m$^{-3}$, with a median of 13.08 $\mu$g m$^{-3}$.

5    Based on Eq. 8, the resulting $N_{INP}^{sea\,spray,air}$ are shown in blue (derived from SML) and green (derived from ULW) in Fig. 8. Irish et al. (2019b) used the same method to get $N_{INP}^{sea\,spray,air}$ in the Arctic (without considering enrichment of INPs in sea salt particles during sea spray generation), as shown by purple (derived from SML) and brown (derived from ULW) boxes in Fig. 8. As discussed in Sect. 3.1, $N_{INP}$ from ULW at Cape Verde are comparable to the Arctic and the NaCl ratios were close to $10^{-10}$ in both studies, therefore, $N_{INP}^{sea\,spray,air}$ (derived from ULW) are also comparable. A high enrichment of $N_{INP}$ in SML to ULW

10   was observed in the Arctic (Irish et al., 2019b). Therefore, $N_{INP}^{sea\,spray,air}$ (derived from SML) in the Arctic was also higher than in this study.

Figure 8 includes $N_{\mathrm{INP}}$ from $PM_{10}$ in this study (shown by black triangles). These values are roughly 4 orders of magnitude above our $N_{\mathrm{INP}}^{\mathrm{sea\,spray,air}}$. But Fig. 8 also shows airborne $N_{\mathrm{INP}}$ as derived for the Southern Ocean (McCluskey et al., 2018a) and the Northeast Atlantic (only clean sector, McCluskey et al., 2018b), which are all above our $N_{\mathrm{INP}}^{\mathrm{sea\,spray,air}}$. As mentioned above, we did not consider a possible enrichment of INPs in SSA compared to the SML or ULW samples. Previous studies

found an enrichment of organic carbon in submicron sea spray particles of about $10^4$ to $10^5$ (Keene et al., 2007; van Pinxteren et al., 2017), and this value decreased to $10^2$ for super-micron particles (Keene et al., 2007; Quinn et al., 2015). It is not clear if INPs are included in the organic carbon for which the enrichment was observed. Also, the INPs we detected in this study were mostly in the super-micron size range. If we increased $N_{\mathrm{INP}}^{\mathrm{sea\,spray,air}}$ by about 2 orders of magnitude in agreement to the enrichment observed for super-micron organic carbon, the resulting $N_{\mathrm{INP}}^{\mathrm{sea\,spray,air}}$ becomes comparable to sea spray INPs

measured in the Southern Ocean (McCluskey et al., 2018a) and the Northeast Atlantic (McCluskey et al., 2018b). But even when considering such an enrichment of INPs, INPs originating from sea spray would only explain a small fraction of all INPs contributing to the measured airborne $N_{\mathrm{INP}}$ in the air at Cape Verde.

## 4 Discussion

$N_{\mathrm{INP}}$ close to sea and cloud level height were compared. One major point of interest is to know whether ground-based mea-
surements can be used to infer aerosol properties at the cloud level. In this study, we found that $N_{\mathrm{INP}}$ are quite similar close to sea level (CVAO) and cloud level (MV) during non-cloud events. But it should still be noted that we only have a small number of filter samples representing non-cloud events in this study. During the observed cloud events, most INPs at MV are activated to cloud droplets. The above findings are in line with what was discussed in the companion paper (Gong et al., 2019b), i.e., (1) the marine boundary is often well mixed at Cape Verde and PNSDs and $N_{\mathrm{CCN}}$ are similar close to both sea and cloud level; (2)
during cloud events, larger particles are activated to cloud droplets.

Most INPs are in the super-micron size range at Cape Verde. We found that about 70% of INPs had a diameter of $>1$ $\mu$m at ice activation temperatures between $-10$ and $-20$ °C. Mason et al. (2016) and Creamean et al. (2018) also found that the majority of INPs is in the super-micron size range in the Arctic, in agreement with the results we obtained here.

Above we derived that $N_{\mathrm{INP}}$ contributed from SSA only accounted for a minor fraction of total $N_{\mathrm{INP}}$ in the air, as well as in
the cloud water at Cape Verde. This still holds even when considering a possible enrichment of INPs in SSA up to $10^2$, which is an enrichment as given in literature for super-micron organic particles (Keene et al., 2007; Quinn et al., 2015). On the other hand, mineral dust is associated with a factor of 1000 higher ice surface site density (a measure to describe the ice activity per particle surface area), compared to SSA (Niemand et al., 2012; DeMott et al., 2016; McCluskey et al., 2018a). In our study, the super-micron particles that make up a large fraction of the INPs we observed were mainly mineral dust, as described in the
accompanying study (Gong et al., 2019b). The comparably high ice activity of super-micron mineral dust and the presence of mainly dust particles in the super-micron size range in our study again supports that indeed most INPs observed in this study were not from sea spray. This is in line with results from Si et al. (2018) and Irish et al. (2019a), both done in the Arctic, where it was also concluded that SSA only contributed little to the INP population. The commonality of these two studies from the

Arctic and the present study is that land was still close enough so that terrestrial sources can have contributed to the observed INPs.

While the above arguments suggest that INPs in our study were mostly mineral dust particles, there were also some measurements with comparably high INP concentrations at temperatures of $-10\,^\circ$C and above. Although it cannot be ruled out that desert dust particles might be ice active at such high temperatures, by examining the reaction of some highly ice active samples to heating, described in Sect. 3.2.1, we found that the most highly ice active INPs on these samples were biological particles. It is an open question where these biological INPs originated. The times during which these highly ice active INPs were observed were times when air masses came from Southern Europe, traveling along the African coast and meanwhile crossing over the region of the Canary Islands. Therefore, for these specific samples, a contribution of INPs from these land sources might be assumed.

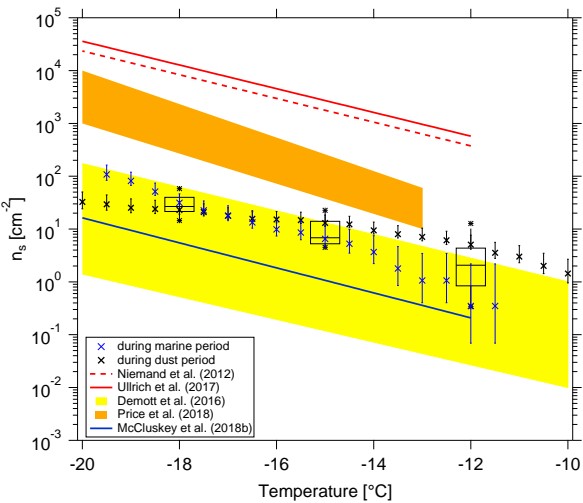

**Figure 9.** Cumulative $n_s$ as a function of temperature in this study is shown by black boxes. The boxes represent the interquartile range. Whiskers represent 10th to 90th percentile. Data not included between the whiskers are plotted as an outlier with a star. Two $n_s$ parameterizations (Niemand et al., 2012; Ullrich et al., 2017) for pure desert dust are shown in dashed and solid red lines, respectively. $n_s$ parameterizations from McCluskey et al. (2018b) for pristine SSA over the Northeast Atlantic are shown as a solid blue line. We also compare to recent data from airborne measurement in a dust layer by Price et al. (2018) in brown shadow and from nascent laboratory generated and ambient SSA by DeMott et al. (2016) in yellow shadow, respectively. $n_s$ during the most clean marine (CVAO 1585) and most dusty (CVAO 1591) periods are shown as blue and black crosses, respectively.

In the following, we will compare $n_s$ derived from our data with that from literature. In Fig. 9, we show the surface site density derived for $N_{\mathrm{INP}}$ from CVAO $PM_{10}$ filters (as shown by black boxes) following Niemand et al. (2012) (details on the surface area are given in the supplement, Fig. S14), together with parameterizations for $n_s$ given by Niemand et al. (2012), Ullrich et al. (2017) and McCluskey et al. (2018b), and the measured $n_s$ given by DeMott et al. (2016) and Price et al. (2018). Niemand et al. (2012) derived $n_s$ from a laboratory study, based on aerosol consisting purely of desert dust particles. It is

therefore reasonable that these mineral dust related $n_s$ values are the largest values shown in Fig. 9, as they are purely related to the mineral dust surface area of an aerosol. All other values shown in Fig. 9 were derived for atmospheric measurements, and the surface area used to derive $n_s$ was always based on measured particle number concentrations. Price et al. (2018) carried out airborne measurements in dust laden air over the tropical Atlantic. Parameterizations from McCluskey et al. (2018b) were

done for pristine SSA over the Northeast Atlantic and both laboratory and atmospheric measurements of SSA were the base for the $n_s$ parameterization given in DeMott et al. (2016). These available $n_s$ parameterizations from previous literature may not be representative for Cape Verde, but we will still compare with them here. $n_s$ derived for our study coincides with the upper range of parameterizations that are otherwise reported for SSA but are clearly lower than values reported for atmospheric desert dust aerosol. This is striking since, as discussed above, INPs observed in this study most likely do not originate from sea

spray, but are dominated by super-micron dust and/or biological particles.

CVAO is a place where marine and dust particles strongly intersect, and both particle types contribute to the surface area. In the companion paper, we have classified the aerosol at CVAO into four different types. Here, in addition to looking at average values as presented above, we selected the most clean marine (CVAO 1585) and most dusty (CVAO 1591) samples for a separate calculation of $n_s$ and added the results to Fig. 9. The $n_s$ is clearly higher for the sample collected during the

dusty period than during the marine period at higher temperatures (roughly $>-16$ °C). However, at temperatures below $-18$ °C it is the other way around. In general, results for these vastly different cases are both still close to the upper limit of the parameterizations reported for SSA.

These comparisons to literature raise the question if and how $n_s$ should be used to parameterize atmospheric INP measurements, which, however, is a question far too prominent to be answered in this study. In general, it is still an open issue to

which extent $N_{INP}$ can be parameterized, based on one or a few parameters, to reliably describe $N_{INP}$ for different locations around the globe. It might prove necessary to develop separate parameterizations for different locations or air masses, as it was already started for parameterizations based on particle number concentrations (see e.g., DeMott et al. (2010); Tobo et al. (2013); DeMott et al. (2015)).

## 5    Summary and conclusions

The MarParCloud campaign took place in September and October 2018 on the island of Cape Verde to investigate aerosols prevailing in the Atlantic Ocean. In addition to a thorough analysis of the atmospheric aerosol particles and CCN in a companion paper (Gong et al., 2019b), samples collected for INPs analysis in this study include: sea surface microlayer (SML) and underlying water (ULW) from the ocean upwind of the island; quartz fiber filter samples of atmospheric aerosol, collected on a tower installed at the island shore and on a 744 m high mountaintop, as well as cloud water collected during cloud events

on the mountaintop. $N_{INP}$ were measured offline with two types of freezing devices, yielding results in the temperature range from roughly $-5$ to $-25$ °C.

Both enrichment and depletion of $N_{\mathrm{INP}}$ in SML to ULW were observed. The enrichment factors (EF) varied from 0.36 to 11.40 and from 0.36 to 7.11 at $-15$ and $-20$ °C, respectively, and were generally independent of the freezing temperature at which $N_{\mathrm{INP}}$ was determined in the freezing devices.

A few CVAO $PM_{10}$ filter samples (CVAO 1596, CVAO 1641 and CVAO 1643) showed elevated $N_{\mathrm{INP}}$ at high temperatures,
e.g., above 0.01 $L^{-1}$ at $-10$°C. These elevated values disappeared after heating the samples at 95 °C for 1 hour. Therefore, biological particles appear to contribute to INPs at these moderate supercooling temperatures. About 83±22%, 67±18% and 77±14% (median±standard deviation) of INPs had a diameter >1 $\mu$m at ice activation temperatures of $-12$, $-15$, and $-18$ °C, respectively, and over the whole examined temperature range, on average roughly 70% of all INPs were super-micron, independent of the temperature. The highly ice active INPs were not found on the CVAO $PM_1$ filters, which suggests that most
of these likely biological INPs are in the super-micron size range.

As MV was in clouds most of the time, only two filters could be collected on MV that were affected by cloud for less than 10% of the sampling time. For these, $N_{\mathrm{INP}}$ were similar at CVAO and MV. During cloud events, most INPs at MV were activated into cloud droplets. These findings aligned very well with the companion paper, i.e., during non-cloud events, PNSDs and $N_{\mathrm{CCN}}$ are similar at CVAO and MV, while during cloud events, larger particles at MV are activated to clouds (see Fig. 8
in the companion paper). When highly ice active particles were present on CVAO $PM_{10}$ filters, they were not observed on MV $PM_{10}$ filters, but were instead observed in the respective cloud water samples. This shows that these INPs are activated into cloud droplets during cloud events.

By comparing $N_{\mathrm{INP}}$ derived for the different examined samples, it was found that values in air and in cloud water agreed well. We also compared atmospheric $N_{\mathrm{INP}}$ to those in SML and ULW, based on the ratio of sodium chloride concentrations
measured for the atmosphere and for SML and ULW. From that we concluded that marine INPs from sea spray can only explain a small fraction of all atmospheric INPs at Cape Verde, unless there would be an enrichment of INPs from SML to the atmosphere by at least a factor of $10^4$. Such an enrichment, however, is higher than anything observed for organic compounds in super-micron particles so far. Summarizing, it can be assumed that most atmospheric INPs detected in the present study were mainly contributed by the dust particles at cold temperatures possibly with few contributions from biological particles at
warmer temperatures.

*Data availability.* The data are available through the World Data Center PANGAEA (https://doi.pangaea.de/10.1594/PANGAEA.906946). A link to the data can be found under this paper's assets tab on ACP's journal website.

*Author contributions.* X. Gong wrote the manuscript with contributions from H. Wex and M. van Pinxteren. C. Stolle, N. Triesch and B. Robinson collected ocean water samples. X. Gong, M. van Pinxteren and N. Triesch collected filter samples. K. W. Fomba collected cloud
water samples. X. Gong and J. Lubitz performed INP measurements. X. Gong preformed data evaluation. X. Gong, H. Wex and F. Stratmann discussed the results and further analysis after the campaign. All co-authors proofread and commented the manuscript.

*Competing interests.* The authors declare that they have no conflict of interests.

*Acknowledgements.* The works were carried out in the framework of the MarParCloud project. The authors acknowledge the Leibniz Association SAW funding for the project "Marine biological production, organic aerosol particles and marine clouds: a Process Chain (MarParCloud)", SAW-2016-TROPOS-2.

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
