# Peer review of "Characterization of aerosol particles at Cape Verde close to sea and cloud level heights - Part 2: ice nucleating particles in air, cloud and seawater"

_Atmospheric Chemistry and Physics, 2019_

## Referee Comment (RC1) · Paul DeMott (Referee) · 15 Oct 2019

**General Comments**

This is a very well executed, and fairly comprehensive study of ice nucleating particles in the Cape Verde region, especially novel in including measurements in all water and air compartments, and attempting to relate these meaningfully.

Overall I had only few comments of significance, and the rest are mostly editorial notes. Specific questions/comments for addressing before publication are listed below.

**Specific Comments**

1) Abstract, last sentence: I struggled to understand this sentence, although I think it is saying that unless there is an unusual SSA INP emission mechanism in the study area, the INPs cannot be from SSA. But the use of the phrase "unless there would be" seems to beg a question that I thought the papered endeavored to answer. As I read, it seems inconceivable. Or are you referring to situations that you did not measure? I suggest thinking about rewording this sentence.

2) Intro, page 2, line 8: A minor note, since it is not relevant for the main topic of this paper. It is difficult to encapsulate this discussion that seems to be required for every INP paper, but this statement does not reflect any role for INP at temperatures lower than -38 °C, which is not the case.

3) Intro, page 2, line 28: Higher than ambient INP concentrations at ground level?

4) Intro, page 2, line 29: I do not understand the meaning of, nor see the need for, the ending phrase of this sentence (i.e., …from the biosphere). It is clear that most INPs come from the biosphere, and the ocean source comes in the form of sea spray aerosol emissions. I favor being explicit.

5) Intro, page 2, line 33: Bigg suggested that INPs were contributed to at least some extent from marine emissions, in the data collected in that region at that time. His abstract statement reads that it is not feasible that they are "only of continental origin."

6) Intro, page 3, lines 10-11: This is an oft-misinterpreted point. These papers parameterize INPs following this segment of the aerosol population, especially in the free troposphere, but the intention is to reflect INPs at all sizes. It is simply a hook to these concentrations, not intended to represent an actual "fraction" of them.

7) Intro, page 3, lines 16: Perhaps add qualifier that these observations were in the Arctic "boundary layer". I bring this up a few times because many things could differ in the free troposphere and at the level of some colder clouds.

[Figure]

8) Experiment and Methods, page 4, line 13: The filter sampler was truly sitting at ground level at MV? Or what was the elevation of the sample head? E.g., 1 m above ground?

9) Experiment and Methods, page 5, line 19: Notes on Table S2. The volumes listed do not seem to work out with other information provided, and the header units seem wrong. First, is duration actually in minutes instead of hours? Also, is std volume (must be L, not $L^{-1}$) for the 2 $cm^2$ surface stated as taken? It does not quite make sense, since if this represented 1/100 of the volume flow of 500 lpm, then about 10 times this volume should have been represented. Please fix header units at least. And state in table description if volume is for the "punch" or the total filter. I will revisit this in the next comment.

10) Experiment and Methods, page 6, lines 24-25: So if I have it correct, this amounts to about 0.75 $cm^2$ of the filter surface area (96, 1 mm punches). Yet this area is stated as 2 $cm^2$ in the previous section. And these new numbers still do not seem to work out to give the sample volumes stated in the tables. I ask these things only if someone wanted to reproduce such work.

11) Experiment and Methods, Section 2.3.2, page 7: Was background testing also done for the other collections, for example by rinsing clean plates use for microlayer sampling and by rinsing the bottles used for seawater collection? Just looking for a few words.

12) ) Experiment and Methods, Section 2.3.3, page 8: I am curious if this calculation of freezing point depression was checked for validation, by for example diluting a seawater sample?

13) Experiment and Methods, Section 2.4, page 8: When introducing surface active site density (the terminology I am used to it being referred to as), it could be good to mention already the fact that when applying it to the total aerosol distribution, this artificially assumes that all particles are the same INP type in the contained surface

area. This is distinct from a laboratory scenario of generating a specific aerosol, and so will fold in all of the influences present in ambient air.

14) Results, page 13, lines 5-6: If bio-INPs, the size range is a bit unexpected for marine bacteria, which tend toward micron or less sizes I think. This might support that the bioaerosols are coming from long distance. It also strikes me that a mention for future work might be to include a collection at >2.5 microns, as one wonders about details of the INP size distribution.

15) Results, page 13, line 12 and Fig. 6: It is not optimal that clouds impacted most of the filters without control, for example by shutting off the pumps, though one understands that clouds are likely pervasive there. This is interpreted as INPs being captured into cloud droplets, and this is supported, but not fully clear. I was struck in Figure 6 by the fact that the CVAO INP concentrations in 6b all appear higher during cloudy periods in comparison to the couple of periods in Fig. 6a with fewer clouds. Any ideas on why this is so? The elevation of INP concentrations is over an order of magnitude in a few cases with the largest humps. This is unusual, but also makes me ask if the conclusions about capture of INP into cloud droplets is the full reason for differences below and within cloud. One factor could be drizzle and precipitation. It should be mentioned are solid if clouds were clearly not drizzling, as this could remove or redistribute INPs.

16) Results, page 14, line 1: Do you discuss cloud diameters in that companion paper? If not, is it consistent with some inference to cloud droplet distributions? This seems to require statement at this point, not later only on page 16.

17) Results, page 16, lines 21-24: It is not stated explicitly, but it seems clear that for this study, the clouds tended toward relatively high water content for marine Cu, with the lowest values equivalent to the assumptions of Petters and Wright, and the highest values exceeding Rangno and Hobbs. This is just a comment. Drop sizing or LWC measurement would be quite useful in any future studies of this type.

18) No reply needed, just to note that the agreement shown in Fig. 8 is striking, even

Interactive
comment

stunning. Also, the discussion at the end of the Results section regarding Fig. 9 is excellent.

19) Discussion, page 20, lines 1-2: The referenced results for the lack of an influence of SSA on INPs is for a completely different location. I am not sure of the relevance of comparing the two studies other than to note that they concluded the same thing. Are you trying to say the SSA INPs never dominate? I would reject that notion. The only commonality in the two regions is that land sources are present at distances within a day or two trajectory distance. This is not the case everywhere.

20) Discussion, page 20, lines 3-4: How does one know what maximum T desert dust is active? Doesn't this study suggests that -10C is not unreasonable?

21) Discussion, page 21, lines 5-6: Is this too prominent a question? Perhaps, but perhaps not. In regions where marine and dust populations strongly intersect, and both populations contribute to the surface area, it seems that it will ultimately be necessary to parse out the contributions. This was not done in DeMott et al. (2016), and that probably makes inclusion of those data as purely SSA somewhat suspect for the data collected in the Caribbean, especially. It makes it difficult to discern anywhere, if a few percent of dust by number is sometimes present (a few papers on this are in press). Do you have any compositional inferences to use here? Consider figures 4 and 5 for of Gong et al. (2019a) for varied compositions during different times. Do the numbers roughly work out if you assume something to use as pure dust? You do not really dig into this at all. It may be a major question, but you appear to have some additional information that would allow you to state if your data are consistent with the proportions of mineral and marine particles.

22) Summary and Conclusions, page 21, lines 19-20: The sentence is not complete, and it is unclear if it is referring to other studies or this one. If referring to this study, I suggest to say that "biological particles appeared to contribute..." I note though that no confirmatory tests were performed to ascertain biological INP influence.

---

## Referee Comment (RC2) · Anonymous Referee #2 · 16 Oct 2019

**Referee report on "Characterization of aerosol particles at Cape Verde close to sea and cloud level heights - Part 2: ice nucleating particles in air, cloud and seawater" by Xianda Gong et al.**

The manuscript presents an analysis of filter and water samples for their content of ice nucleating particles (INPs). Samples were collected at the Cabo Verde Island as part of the MarParCloud project. What makes this study unique is that samples were simultaneously taken at sea and cloud level, and include filter samples of ambient air as well as cloud water and sea water samples. By comparing the INP content in these different compartments of the environment the authors infer the contribution of particles of certain origin and characteristics to the atmospheric INP population.

There are several points that need to be improved -- these are listed in detail below. Once these are addressed, the paper should be published in ACP.

Specific comments

1) P.1 ln.14f Attributing the difference in $N_{INP}$ from PM10 and PM1 samples to biological particles only based on the temperature of activation is speculative and over-reaching, eg. super-micron mineral dust particles can also cause ice formation at -10°C (see Hoose and Möhler, 2012). Are there indications against mineral dust particles being responsible for the higher activity? Were elevated $N_{INP}$ observed during dust events?

2) P.1 ln.18 Same as above. Most particles >1um are probably activated as CCN. There is no evidence for a biological origin of these INPs.

3) P.2 ln.7 In addition to homogeneous ice nucleation, heterogeneous ice nucleation by deposition nucleation is active also below -38°C. As the sentence is written it indicates that only homogeneous nucleation would be active below -38°C, which is incorrect.

4) P.2 ln.9 Are all droplets aqueous solutions?

5) P.2 ln.12 If there is a special aspect to the -20°C reported in Augustin-Bauditz et al., 2014 it should be mentioned. If not, the "below -15°C" from Hoose and Möhler, 2012 already includes -20°C and the Augustin-Bauditz et al., 2014 reference can be omitted. According to Fig. 3d in Hoose and Möhler, 2012, super-micron mineral dust particles can be active INP already below -10°C and not -15°C.

6) P.2 ln.17 The results in Boose et al., 2016 suggest desert dust to nucleate ice mostly below -25°C, especially airborne samples and the study shows differences among samples from different regions. This seems to contradict what is stated in ln.12-16. This section of the introduction should be revised, distinguishing mineral dust from desert dust studies and motivating the relevance of desert dust for the results presented here.

7) P.2 ln.26 Specify what is meant by "differences in desert sources".

8) P.2 ln.28 Is $N_{INP}$ at ground level lower because the dust loading was lower at ground level?

9) P.2 ln.32f This is incorrect. Bigg, 1973 suggested that INPs "are transported from a distant land source, or from a stratospheric source, and brought to sea level by convective mixing." Schnell and Vali, 1976 suggested a marine source could explain the observations of Bigg 1973.

10) P.3 ln.2 What further evidence? This sentence seems to refer in a more general tone to the same study as the sentence before. Please clarify.

11) P.3 ln.10f DeMott et al. 2010, 2015 suggest a correlation of $N_{INP}$ and the concentration of particles above a certain size. However they do not specify that only a fraction of large particles would act as INPs. Double check.

12) P.3 ln.20 The deduction that most biological INPs occur together with their original carrier appears incomplete. Clarify.

13) P.4 first paragraph of Sec. 2.1.1. If the CVAO station is located on the northwest shore (ln. 11) of the island and wind direction is from the northeast (ln. 17), does this mean air first crossed the island before reaching the station? I think the CVAO is located on the northeast shore.

14) P.5 ln.16 It is mentioned in Sec. 2.1.1. that measurements took place during 31 days. Why where only 17 or 19 filters collected? How long was the sampling period per filter?

15) P.6 ln.29 By "brightness change" do you mean a change in transmitted light due to a change in opacity when droplets freeze?

16) P.7 ln.8f Can you explain how the assumption of a Poisson distribution influences the result of Eq.1?

17) P.7 sec.2.3.1. The principle of determining the cumulative INP concentration as well as the difference between INDA and LINA experiments are not very well explained. The explanation should mention that all wells contain multiple INPs but only the most active one (active at the highest temperature) causes freezing. The probability to find an active INP at a certain temperature increases with sample volume.

18) P.7 Sec.2.3.2 Provide the equations you used for this calculation and some values to inform the reader how large statistical error and background are.

19) P.7 ln.18 It is not explained previously that washing water was used or how it was prepared.

20) P.7 ln.18 How was the information about the number of INPs per well obtained?

21) P.7 ln.20 A short explanation of the method from Agresti and Coull, 1998 would be helpful here.

22) P.8 ln.3f Can you provide a range of the derived freezing point depression for the samples. Was the freezing point depression experimentally confirmed, eg. by measuring the melting point depression?

23) P.8 ln.9 The description of $n_s$ is imprecise. $N_s$ gives the number of ice active sites per surface area, here the surface of all aerosol, ice active or not. Revise.

24) P.8 ln.20-26 This section is speculative and it is not clear why this comparison is relevant for the present study. Clarify.

25) P.8 ln.27 Add an introducing sentence mentioning that the following analysis is done to compare $N_{INP}$ found in SML and ULW at Cabo Verde and explain why enrichment/depletion could be expected. Could INP in the SML originate from settling aerosol?

26) P.9 Fig.1 Why were samples 6, 7, 8, 10 and 11 not used? Instead of comparing to Wilson's data from a different environment, a direct comparison of SML to ULW by plotting both data on top of each other might show more clearly that $N_{INP}$ are the same between the two.

27) P.9 ln.2 Clarify that you refer to the temperature at which $N_{INP}$ was determined in the drop freezing experiment. As is, it could be misunderstood as water temperature during sampling.

28) P.9 ln.3 Can you provide an explanation for the variation in EF with temperature? Does the interpretation change when considering the confidence interval of $N_{INP}$? Do an error propagation of Eq. (5) and estimate the error in EF (should be included in Fig.2).

29) P.9 ln.8 Shouldn't SML thickness be related to concentration of dissolved organic matter? Explain why SML at Cabo Verde is larger than the SML in the Wilson et al., 2015 study even though conditions are oligotrophic.

30) P.10 Fig.2 add error estimation of EF.

31) P.10 ln.9-11 What could be the reason for the variation between samples and why is the range of variation consistent to measurements at other locations? Could the number of samples or sampling duration determine the range of variation?

32) P.10 ln.12f Testing the heat sensitivity of the 3 samples could substantiate the interpretation that biological particles are responsible for the enhanced $N_{INP}$.

33) P.10 ln.14ff. This paragraph is difficult to follow. Do you mean above -16.8°C concentrations within 2°C of each other are correlated? Looking at the data in Fig.3, $N_{INP}$ seem to change very little in 0.1°C steps, but are different between individual measurements. What is the actual regression model for which you report $R^2$ and p value? What data was used for the regression? Check the statistical power of correlating only few data points. Looking at the data in a differential spectrum (see Vali, 1971) might be a better method to identify temperatures where INPs of different origin become active.

34) P.11 ln.5 The dataset is by far not large enough to construct a robust pdf. The result in Fig.4 is vastly dependent on the choice of intervals to bin the data. The given pdf is therefore not suitable to perform any data analysis as the result could depend on the binning of data.

35) P.12 Fig.4 I suggest to remove Fig.4. It is not illustrating new information that is not already contained in Fig.3 and Fig.5. If you chose to keep Fig.4 check y-axis, the area under the pdf should be 1 or 100%.

36) P.12 ln.6ff Explain how the values in this paragraph were derived. I assume you compare the $N_{INP,PM10}$ to $N_{INP,PM1}$ from filters collected during the same time period and take the ratio?

37) P.13 ln.3ff Last sentence of this paragraph is speculative and repeating for the 6[th] time in this manuscript that high temperature activity of PM10 filter samples could be due to biological particles. As this seems to be a central point in your interpretation of the data I strongly recommend to experimentally test the heat sensitivity of $N_{INP}$ (eg. following the procedure described in Joly et al., 2014) to support that biological particles are causing the mentioned difference.

38) P.14 ln.1 The difference in $N_{INP}$ (shown in Fig.6 (b)) is clearly visible above -20°C, not only above -17°C.

39) P.14 ln.3f Is there evidence for a substantial fraction of droplets below 10um? Even though no direct observations are available from MV, observations in similar environments could help this discussion. Measurements of orographic cloud droplet distributions e.g. from Hawaii showed a bimodal droplet size spectra with both modes >10um (Squires, 1958).

40) P.14 ln.5 This speculation is repeated several times throughout the paper but no evidence to support the biological nature of these INPs is presented. Either conduct heat sensitivity experiments and/or provide electron microscope images of large biological particles on the filters to demonstrate that this is a plausible interpretation, or delete the statement.

41) P.15 ln.4ff In some cases over 100% difference in $Na^+$ and $Cl^-$ concentration between the present study and Gioda et al., 2009 seem large and not comparable. In contrast to what other values do the authors think concentrations are comparable?

42) P.16 ln.13ff Couldn't $F_{cloud\_air}$ be estimated directly from the water collection rate of the CASCC2? This would reduce the uncertainty for the estimation of $N_{INP}$.

43) P.17 ln.1f Does this range include the error estimation from the INP experiment? The two uncertainties (in $F_{cloud\_air}$ and $N_{INP}$ in cloud water) should be combined by error propagation when deriving the range of $N_{INP,air}$.

44) P.17 ln.3 The uncertainty range spreads over 2 orders of magnitude while the $N_{INP}$ cover 4 orders of magnitude. "general agreement" seems to have limited meaning here. The sensitivity on $d_{drop}$ when calculating $N_{INP,air}$ from water samples determines the result. As already suggested above, could the amount of collected cloud water be used to determine $F_{cloud\_air}$ or to constrain the $d_{drop}$ range? Alternatively LWC can be estimated from the CASCC2 collection rate (Sec. 2.3. in Demoz, 1996) for different drop size distributions (that could

come from the literature eg. Sqires, 1958). Another option to estimate LWC might be to use the NaCl content in cloud water and air as a tracer, similar to the method applied in Sec.3.4.

45) P.17 ln.5 Instead of the collection efficiency at 3.5 um, it would be more useful to know the collection efficiency above the cut-off of the PM10 inlet of the filter sampler, where the droplet fraction not collected by the filter sampler but the cloud water collector should be found. According to Demoz, 1996 collection efficiency above 10um should be >80%. The high collection efficiency above 10um does not support the given explanation for a difference in $N_{INP}$ from filter and cloud water samples which is provided at the beginning of the paragraph (ln.3ff). Revise.

46) P.18 Eq. (7) and Fig.9  An error estimation for $N_{INP}$ from sea spray by error propagation of input variables in Eq. 7. Include error estimate in Fig.9.

47) P.18 ln.7 Did you use the individually measured NaCl concentration for each sampling period or the median to calculate the INP concentration in air? What is the range of the NaCl ratio on the right hand side of Eq.7?

48) P.18 ln.10f Related to the previous comment. $N_{INP}$ at Cabo Verde and in the Arctic should be the same only if the NaCl ratio in Eq.7 is also the same. Alternatively, this highlights that the result of Eq.7 is largely insensitive to the NaCl ratio. This should be clarified.

49) P.19 ln.1-12 It is unclear why enrichment of OC in SML is discussed here as no connection to $N_{INP}$ has been established. I recommend to delete this paragraph. All that can be said is that airborne $N_{INP}$ are higher than whatever $N_{INP}$ could have originated from the ocean.

50)  P.20 Fig.10 I suggest to include $n_s$ for all temperatures covered by your experiments and for filter, water, CVOA, MV separately.

51) P.20 ln.9 Fig.10 should be motivated by stating what the expected $n_s$ are (SSA or dust) and then argue that the available $n_s$ parametrizations are not representative for Cabo Verde. It might be not surprising that $n_s$ parametrizations that are based on measurements in other environments do not capture the situation at Cabo Verde. Additionally, specify which data of "our data" is shown in Fig.10. Is it filter, water, CVAO or MV? As suggested in the previous comment all of these datasets could be of interest.

52) P.21 ln.5f Here, the authors could suggest future directions, eg. regional, seasonal $n_s$ parametrizations or parameterizing $N_{INP}$ directly from field observations without employing surface area specific activity.

53) P.21 ln.19f Repetition of sentence from p.10 ln.12f. What could be the origin of these super-micron biological particles? Doesn't the evidence in this paper rather point to super-micron mineral dust?

54) P.21 ln.24 Provide evidence for biological particles or include dust as a possible source.

55) P.21 ln 26-27 Either, add figures showing both $N_{CCN}$ and PNSD at the two locations during cloud events and non-cloud events and provide difference in PNSD where the CCN active INPs are found, or, point the reader to Fig.8 in the companion paper. Was there a trend in $N_{INP}$ related to the particle types indicated in Fig.8 of the companion paper?

Technical corrections

1) p.1 ln.4f SML and ULW might be sources of INPs, but sea and cloud level are compartments of the atmosphere where $N_{INP}$ are measured, not sources. Rephrase.

2) P.1 ln.7 When mentioning "temperature" be specific in which system the temperature was measured. Here: "trends of EF with temperature." Temperature of what? Sea water, ambient or in the INP experiment?

3) P.1 ln.8 Same as above. "at any particular temperature" could be understood as if sampling was conducted at different temperatures. The authors should be more specific and say: the temperature to which samples were exposed to in ice nucleation experiment.

4) P.2 ln.10 Freezing is not the same as ice nucleation. Immersion freezing refers to an ice nucleation mechanism rather than the freezing process. Rephrase.

5) P.2 ln.15 Replace "more effective" with "more active" instead.

6) P.2 ln.22 Do you mean North African desert?

7) P.2 ln.27 Replace "ice nucleating properties" by "$N_{INP}$"

8) P.3 ln.1 Replace "INPs" with "the ice nucleation activity".

9) P.3. ln.22 Add: assuming that most INPs activate as CCN.

10) P.3 ln.23 Specify: "in rain samples"

11) P.3 ln.31 Replace "for INPs analysis" with "to measure $N_{INP}$"

12) P.5 ln.4-6. Repetition of "specially designed". Delete in line 4-5.

13) P.5 ln.12 Is there something special about the Digitel filter sampler from the reseller Walter Riemer Messtechnik? If not the manufacturer should be referenced instead.

14) P.5 ln.13 Move (Munktell, MK 360) to after "filters".

15) P.8 ln.12 Please add units to variables.

16) P.8 ln15 Mention that this sentence refers to individual samples and starting from ln.17 variation between the 9 samples is discussed.

17) P.8 ln.26 Instead of "This" start the sentence with "The low biological activity in the SML around Cabo Verde"

18) P.9 Fig.1 Use the same y-axis scale for SML and ULW and include gridlines to facilitate comparing SML to ULW. Consider plotting the data on top of each other.

19) P.9 ln.7 Specify if you refer to sampling, or INP experiment technique.

20) P.9 ln.10 Delete "the"

21) P.10 ln.7 "contribute" instead of "contributes"

22) P.10 ln.7 Replace "few" with "two"

23) P.13 Fig.5 Check unit of y-axis. Should not be %. Add to the figure caption what range is represented by the box and whisker of the boxplot. Due to the limited number of samples it would be better to just provide the range instead of a boxplot (which requires the assumption of an underlying distribution).

24) P.13 ln.12, 14, 17 Avoid vague qualifiers "more or less", "only little", "quite similar", "mostly" and quantify instead.

25) P.13 ln.19 Add: "…obvious from Fig.6 that…"

26) P.14 Fig.6 Add (a) and (b) to the subfigures and add gridlines for easier readability. In the caption put the (a),(b) before describing the subfigure: "… MV $PM_{10}$ filters during (a) less (cloud time fraction <10%) cloud effected periods and (b) highly…"

27) P.16 ln.25-28 Give the equation for this calculation.

28) P.17 Fig.8 First sentence in figure caption is incomplete. Replace "shown by" with "shown as".

29) P.18 Fig.9 Caption: "error bars showing" instead "error bars show"

30) P.19 ln.26-28 Revise structure of sentence.

31) P.19 ln.31 Add: "… ice activity of super-micron mineral dust…"

32) P.20 ln.5 Add: "… associated with biological particles, but has also been observed for super-micron dust samples (Hoose and Möhler, 2012)."

33) P.21 ln.4 You could add: "… do not originate from sea spray, but are dominated by super-micron dust."

34) P.21 ln.9 Instead of "thorough analysis" specify what kind of analysis is shown in the companion paper.
35) P.21 ln.13f Freezing experiments with the devices used for this study should give reliable data up to 0°C and not "roughly" from below -5°C.
36) P.21 ln.15f Revise after correcting Sec. 3.1.
37) P.21 ln.21ff Revise after correcting Sec. 3.2.1.
38) P.21 ln.25 "quite similar" should be put into perspective based on the limited number of investigated samples.
39) P.21 ln.30f Revise after correcting Sec. 3.3.2

References

Agresti, A. and Coull, B. A.: Approximate is Better than ''Exact" for Interval Estimation of Binomial Proportions, The American Statistician, 52, 119–126, https://doi.org/10.1080/00031305.1998.10480550, 1998.

Augustin-Bauditz, S., Wex, H., Kanter, S., Ebert, M., Niedermeier, D., Stolz, F., Prager, A., and Stratmann, F.: The immersion mode ice nucleation behavior of mineral dusts: A comparison of different pure and surface modified dusts, Geophysical Research Letters, 41, 7375–7382, https://doi.org/10.1002/2014gl061317, 2014.

Bigg, E. K.: Ice Nucleus Concentrations in Remote Areas, Journal of the Atmospheric Sciences, 30, 1153–1157, https://doi.org/doi:10.1175/1520-0469(1973)030<1153:INCIRA>2.0.CO;2, 1973.

Boose, Y., Welti, A., Atkinson, J., Ramelli, F., Danielczok, A., Bingemer, H. G., Plötze, M., Sierau, B., Kanji, Z. A., and Lohmann, U.: Heterogeneous ice nucleation on dust particles sourced from nine deserts worldwide – Part 1: Immersion freezing, Atmospheric Chemistry and Physics, 16, 15 075–15 095, https://doi.org/10.5194/acp-16-15075-2016, 2016.

DeMott, P. J., Prenni, A. J., Liu, X., Kreidenweis, S. M., Petters, M. D., Twohy, C. H., Richardson, M. S., Eidhammer, T., and Rogers, D. C.:Predicting global atmospheric ice nuclei distributions and their impacts on climate, Proceedings of the National Academy of Sciences, 107, 11 217–11 222, https://doi.org/10.1073/pnas.0910818107, 2010.

DeMott, P. J., Prenni, A. J., McMeeking, G. R., Sullivan, R. C., Petters, M. D., Tobo, Y., Niemand, M., Möhler, O., Snider, J. R., Wang, Z.,and Kreidenweis, S. M.: Integrating laboratory and field data to quantify the immersion freezing ice nucleation activity of mineral dust particles, Atmos. Chem. Phys., 15, 393–409, https://doi.org/10.5194/acp-15-393-2015, 2015.

Demoz, B. B., Collett, J. L., and Daube, B. C.: On the Caltech Active Strand Cloudwater Collectors, Atmospheric Research, 41, 47–62, https://doi.org/https://doi.org/10.1016/0169-8095(95)00044-5,1996.

Gioda, A., Mayol-Bracero, O. L., Morales-García, F., Collett, J., Decesari, S., Emblico, L., Facchini, M. C., Morales-De Jesús, R. J., Mertes, S., Borrmann, S., Walter, S., and Schneider, J.: Chemical Composition of Cloud Water in the Puerto Rican Tropical Trade Wind Cumuli, Water, Air, and Soil Pollution, 200, 3–14, https://doi.org/10.1007/s11270-008-9888-4, 2009.

Hoose, C. and Möhler, O.: Heterogeneous ice nucleation on atmospheric aerosols: a review of results from laboratory experiments, Atmos.Chem. Phys., 12, 9817–9854, https://doi.org/10.5194/acp-12-9817-2012, 2012.

Joly, M., Amato, P., Deguillaume, L., Monier, M., Hoose, C., and Delort, A. M.: Quantification of ice nuclei active at near 0 °C temperatures in low-altitude clouds at the Puy de Dôme atmospheric station, Atmos. Chem. Phys., 14, 8185–8195, https://doi.org/10.5194/acp-14-8185-2014, 2014.

Schnell, R. and Vali, G.: Freezing nuclei in marine waters, Tellus, 27, 321–323, 1975.

Squires, P.: The Microstructure and Colloidal Stability of Warm Clouds, Tellus, 10:2, 256-261, DOI: 10.3402/tellusa.v10i2.9229, 1958.

Vali, G.: Quantitative Evaluation of Experimental Results an the Heterogeneous Freezing Nucleation of Supercooled Liquids, Journal of the Atmospheric Sciences, 28, 402–409, https://doi.org/10.1175/1520-0469(1971)028<0402:qeoera>2.0.co;2, 1971.

Wilson, T. W., Ladino, L. A., Alpert, P. A., Breckels, M. N., Brooks, I. M., Browse, J., Burrows, S. M., Carslaw, K. S., Huffman, J. A., Judd, C., Kilthau, W. P., Mason, R. H., McFiggans, G., Miller, L. A., Najera, J. J., Polishchuk, E., Rae, S., Schiller, C. L., Si, M., Temprado, J. V., Whale, T. F.,Wong, J. P. S.,Wurl, O., Yakobi-Hancock, J. D., Abbatt, J. P. D., Aller, J. Y., Bertram, A. K., Knopf, D. A., and Murray, B. J.: A marine biogenic source of atmospheric ice-nucleating particles, Nature, 525, 234–238, https://doi.org/10.1038/nature14986, 2015.

---

## Author Comment (AC1) · 4 Dec 2019

Dear Paul,

Thanks for doing this review, for your thorough reading of our manuscript and for your positive comments. The points you raised were mindful and certainly helped improving this manuscript. Below, please find your original comments in blue and our responses in black. When referencing page and line numbers, we are always referring to the new versions of manuscript and SI, which are attached at the end. Concerning the literature cited in this answer to your review, we ask you to refer to the attached new version of the manuscript with tracked changes.

**General Comments**

This is a very well executed, and fairly comprehensive study of ice nucleating particles in the Cape Verde region, especially novel in including measurements in all water and air compartments, and attempting to relate these meaningfully. Overall I had only few comments of significance, and the rest are mostly editorial notes. Specific questions/comments for addressing before publication are listed below.

**Specific Comments**

1) Abstract, last sentence: I struggled to understand this sentence, although I think it is saying that unless there is an unusual SSA INP emission mechanism in the study area, the INPs cannot be from SSA. But the use of the phrase "unless there would be" seems to beg a question that I thought the papered endeavored to answer. As I read, it seems inconceivable. Or are you referring to situations that you did not measure? I suggest thinking about rewording this sentence.

We changed last sentence in the abstract such that this should be clear now:

"This latter conclusion still holds when accounting for an enrichment of organic carbon in super-micron particles during sea spray generation as reported in literature."

2) Intro, page 2, line 8: A minor note, since it is not relevant for the main topic of this paper. It is difficult to encapsulate this discussion that seems to be required for every INP paper, but this statement does not reflect any role for INP at temperatures lower than -38 C, which is not the case.

Thanks for your comment. We changed it as:

"Ice crystals in the atmosphere can be formed either via homogeneous nucleation below −38 °C or via heterogeneous nucleation aided by aerosol particles known as ice nucleating particles (INPs) at any temperature below 0 °C."

3) Intro, page 2, line 28: Higher than ambient INP concentrations at ground level?

Yes, it is. We extended this sentence as:

"Schrod et al. (2017) found that mineral dust or a constituent related to dust was a major contributor to $N_{INP}$ of the aerosol on Cyprus, and $N_{INP}$ in elevated dust plumes was on average a factor of 10 higher than $N_{INP}$ at ground level, where the dust loading is lower."

4) Intro, page 2, line 29: I do not understand the meaning of, nor see the need for, the ending phrase of this sentence (i.e., : : :from the biosphere). It is clear that most INPs come from the biosphere, and the ocean source comes in the form of sea spray aerosol emissions. I favor being explicit.

We deleted this ending phrase "which would…".

5) Intro, page 2, line 33: Bigg suggested that INPs were contributed to at least some extent from marine emissions, in the data collected in that region at that time. His abstract statement reads that it is not feasible that they are "only of continental origin."

You are correct in that Bigg (1973) suggested that the INPs are not only of continental origin, but he does not mention marine sources at all. Instead, he argues that there should be a stratospheric source. As the second review suggested, the text in our manuscript was changed to:

"Based on a long-term measurement of INPs in the marine boundary layer in the south of and around Australia, Bigg (1973) suggested that INPs in ambient air were from a distant land source, or from a stratospheric source, and brought to sea level by convective mixing. Schnell and Vali. (1976) suggested a marine source could explain the observations of Bigg (1973)."

6) Intro, page 3, lines 10-11: This is an oft-misinterpreted point. These papers parameterize INPs following this segment of the aerosol population, especially in the free troposphere, but the intention is to reflect INPs at all sizes. It is simply a hook to these concentrations, not intended to represent an actual "fraction" of them.

Thanks for clarifying this. We reworded this sentence.

"Simultaneous measurements of $N_{INP}$ and particle number size distributions were used to develop parameterizations in which $N_{INP}$ depends on a temperature dependent fraction of all particles with sizes above 500 nm (DeMott et al., 2010, 2015)."

7) Intro, page 3, lines 16: Perhaps add qualifier that these observations were in the Arctic "boundary layer". I bring this up a few times because many things could differ in the free troposphere and at the level of some colder clouds.

It was changed to:

"Creamean et al. (2018) also found that super-micron or coarse mode particles are the most proficient INPs at warmer temperatures in the Arctic boundary layer and they might be biological INPs."

8) Experiment and Methods, page 4, line 13: The filter sampler was truly sitting at ground level at MV? Or what was the elevation of the sample head? E.g., 1 m above ground?

The filter sampling was done using a Digitel filter sampler. The filter head was about 2 m above ground level. As mentioned in the paper, the height of MV is 744 m and the inlet height is 746 m. We added:

"… on the ground with the inlet 2 m above the bottom, …"

9) Experiment and Methods, page 5, line 19: Notes on Table S2. The volumes listed do not seem to work out with other information provided, and the header units seem wrong. First, is duration actually in minutes instead of hours? Also, is std volume (must be L, not L-1) for the 2 cm2 surface stated as taken? It does not quite make sense, since if this represented 1/100 of the volume flow of

500 lpm, then about 10 times this volume should have been represented. Please fix header units at least. And state in table description if volume is for the "punch" or the total filter. I will revisit this in the next comment.

Thanks a lot for discovering these errors.

The "Duration" should be in minutes. The "Total Volume" should be in std m$^3$ and "Volume Per Well" should be in std L. The volume is always given in standard temperature and pressure (0 °C and 1013.25 hPa).

10) Experiment and Methods, page 6, lines 24-25: So if I have it correct, this amounts to about 0.75 cm2 of the filter surface area (96, 1 mm punches). Yet this area is stated as 2 cm2 in the previous section. And these new numbers still do not seem to work out to give the sample volumes stated in the tables. I ask these things only if someone wanted to reproduce such work.

For the here presented measurements, only a small piece of 2 cm in diameter was used, from the much larger filter. From this small piece, then pieces with 1 mm in diameter were punched out. The area of these 1 mm needs to set into relation with the size of the overall sampled area. For the filters with diameters of 150 mm, the effective sampling area had a diameter of 140 mm. We made this clearer in the text:

"on 150 mm in diameter quartz fiber filters (Munktell, MK 360) with an effective sampling area of 140 mm in diameter."

"…a circular piece of these filters of 2 cm in diameter was used from which then smaller pieces were punched out for the analysis (see section 2.2)."

11) Experiment and Methods, Section 2.3.2, page 7: Was background testing also done for the other collections, for example by rinsing clean plates use for microlayer sampling and by rinsing the bottles used for seawater collection? Just looking for a few words.

We tested the MilliQ water at Cape Verde and it was as clean as MilliQ water at TROPOS. But we did not test the MilliQ water after washing these glass plates and the containers in which the samples were stored. But we at least explicitly say now, at the end of this section:

"For those samples that were already collected in a liquid state (ULW, SML and cloud water), a background correction was not done."

12) Experiment and Methods, Section 2.3.3, page 8: I am curious if this calculation of freezing point depression was checked for validation, by for example diluting a seawater sample?

We did not test the freezing point depression previously, but did it now. Since the seawater samples are no longer available, we tested the freezing depression of a pure sodium chloride solution.

We dissolved 0.72 g sodium chloride in 20 mL MilliQ water to get a solution with a salinity similar to that of the SML and ULW samples. The frozen fraction ($f_{ice}$) of MilliQ water and of this sodium chloride solution are shown in the figure below. The error bars show 95% confidence intervals of $f_{ice}$. Due to large measurement uncertainties for the first frozen droplets, the freezing point depression should rather be determined from temperatures below approx. −25 °C, where, indeed, a freezing point depression temperature about ~2.2 °C was observed. It is therefore acceptable to use a freezing point depression of 2.2 °C in this study.

[Figure]

13) Experiment and Methods, Section 2.4, page 8: When introducing surface active site density (the terminology I am used to it being referred to as), it could be good to mention already the fact that when applying it to the total aerosol distribution, this artificially assumes that all particles are the same INP type in the contained surface area. This is distinct from a laboratory scenario of generating a specific aerosol, and so will fold in all of the influences present in ambient air.

We followed your suggestion and extended this part as:

"For cases where a single type of aerosol, such as one type of mineral dust, is examined in laboratory studies, $A_{total}$ can be the total particle surface area. However, when field experiments are done, using the total particle surface area of the atmospheric aerosol assumes that all particles contribute to INP and have the same $n_s$, while the vast majority of these particles will not even be an INP. On the other hand, singling out the contribution of separate INP types in the atmospheric aerosol and relying $n_s$ only to them by using their contribution to the total surface area is at least demanding if not often impossible. This has to be kept in mind when interpreting heterogeneous ice nucleation in terms of $n_s$"

14) Results, page 13, lines 5-6: If bio-INPs, the size range is a bit unexpected for marine bacteria, which tend toward micron or less sizes I think. This might support that the bioaerosols are coming from long distance. It also strikes me that a mention for future work might be to include a collection at >2.5 microns, as one wonders about details of the INP size distribution.

We followed your suggestion and added:

"This suggests that these biological INPs might originate from long-range transport, as marine biological INPs were usually reported to be submicron in size (Wilson et al., 2015, Irish et al.,2017). The contribution of SSA to INPs will be discussed further in section 3.4."

15) Results, page 13, line 12 and Fig. 6: It is not optimal that clouds impacted most of the filters without control, for example by shutting off the pumps, though one understands that clouds are likely pervasive there. This is interpreted as INPs being captured into cloud droplets, and this is supported, but not fully clear. I was struck in Figure 6 by the fact that the CVAO INP

concentrations in 6b all appear higher during cloudy periods in comparison to the couple of periods in Fig. 6a with fewer clouds. Any ideas on why this is so?

There is only a small number of samples. It is seen that during marine events, $N_{INP}$ is lower than during the other times, even up to high ice nucleation temperatures. It is also seen that the cloud free times occurred during the marine times. However, we did not find a good correlation between coarse mode particle number concentration and $N_{INP}$. These are interesting observations, indeed. But as we only have this very low number of samples collected during the marine period, we not expand on this topic in the text.

The elevation of INP concentrations is over an order of magnitude in a few cases with the largest humps. This is unusual, but also makes me ask if the conclusions about capture of INP into cloud droplets is the full reason for differences below and within cloud. One factor could be drizzle and precipitation. It should be mentioned are solid if clouds were clearly not drizzling, as this could remove or redistribute INPs.

In the companion paper, we characterized the cloud events at MV by comparing the PNSDs at CVAO and MV. The cloud events are pervasive at MV. During the cloud events, most accumulation mode and partly also Aitken mode particles were activated to cloud droplets. We did not have a functioning APS at MV, therefore we cannot compare the coarse mode particles. However, we can assume that most coarse mode particles were activated to cloud droplets because larger particle are easier to activate. INPs are mainly in the super-micron size range, which can explain why most INPs were captured by cloud. There was no precipitation observed at the foothill, which means cloud droplets instead of drizzling were present on the mountaintop.

16) Results, page 14, line 1: Do you discuss cloud diameters in that companion paper? If not, is it consistent with some inference to cloud droplet distributions? This seems to require statement at this point, not later only on page 16.

We did not discuss cloud droplet diameters in the companion paper. But we now added the following in page 15 lines 4-6:

"These observations are consistent with results by Siebert and Shaw (2017) who observed broad cloud droplet size distributions in a size range from ~ 5 to 25 μm in shallow cumulus clouds, with the maximum of the distribution still being below 10 μm."

17) Results, page 16, lines 21-24: It is not stated explicitly, but it seems clear that for this study, the clouds tended toward relatively high water content for marine Cu, with the lowest values equivalent to the assumptions of Petters and Wright, and the highest values exceeding Rangno and Hobbs. This is just a comment. Drop sizing or LWC measurement would be quite useful in any future studies of this type.

In this study, the LWC was calculated from Equation 6. In this function, we assumed droplet size varies between 7 and 20, with a median value of 15 μm. Our calculation of the LWC is indeed sensitive to the droplet size. However, with the assumption of a range from 7 to 20 μm, we can safely assume to cover all possible values of the LWC. As you mentioned, measuring droplet size or LWC would be very helpful in these kinds of study.

18) No reply needed, just to note that the agreement shown in Fig. 8 is striking, even stunning. Also, the discussion at the end of the Results section regarding Fig. 9 is excellent.

Thanks for saying that!

19) Discussion, page 20, lines 1-2: The referenced results for the lack of an influence of SSA on INPs is for a completely different location. I am not sure of the relevance of comparing the two studies other than to note that they concluded the same thing. Are you trying to say the SSA INPs never dominate? I would reject that notion. The only commonality in the two regions is that land sources are present at distances within a day or two trajectory distance. This is not the case everywhere.

The sentence might have been misleading, as it was not meant to say that SSA INPs never dominate. We changed it such that this should be clear now:

"This is in line with results from Si et al. (2018) and Irish et al. (2019a), both done in the Arctic, where it was also concluded that SSA only contributed little to the INP population. The commonality of these two studies from the Arctic and the present study is that land was still close enough so that terrestrial sources can have contributed to the observed INPs."

20) Discussion, page 20, lines 3-4: How does one know what maximum T desert dust is active? Doesn't this study suggests that -10C is not unreasonable?

You are correct in that there is no maximum T for the activity of desert dust particles. But we also cannot claim the opposite. As the second review suggested, samples CVAO 1596, CVAO 1641 and CVAO 1643 were heated to 95 °C for 1 hour and a large reduction of $N_{INP}$ was observed. Therefore, this paragraph was changed to:

"While the above arguments suggest that INPs in our study were mostly mineral dust particles, there were also some measurements with comparably high INP concentrations at temperatures of $-10\,°C$ and above. Although it cannot be ruled out that desert dust particles might be ice active at such high temperatures, by examining the reaction of some highly ice active samples to heating, described in Sec. 3.2.1, we found that the most highly ice active INPs on these samples were biological particles. It is an open question where these biological INPs originated. The times during which these highly ice active INPs were observed were times when air masses came from Southern Europe, travelling along the African coast and meanwhile crossing over the region of the Canary Islands. Therefore, for these specific samples, a contribution of INPs from these land sources might be assumed. "

21) Discussion, page 21, lines 5-6: Is this too prominent a question? Perhaps, but perhaps not. In regions where marine and dust populations strongly intersect, and both populations contribute to the surface area, it seems that it will ultimately be necessary to parse out the contributions. This was not done in DeMott et al. (2016), and that probably makes inclusion of those data as purely SSA somewhat suspect for the data collected in the Caribbean, especially. It makes it difficult to discern anywhere, if a few percent of dust by number is sometimes present (a few papers on this are in press). Do you have any compositional inferences to use here? Consider figures 4 and 5 for

of Gong et al. (2019a) for varied compositions during different times. Do the numbers roughly work out if you assume something to use as pure dust? You do not really dig into this at all. It may be a major question, but you appear to have some additional information that would allow you to state if your data are consistent with the proportions of mineral and marine particles.

Thanks for your comment. It is really interesting for the future work to parse out the SSA and dust contribution to INPs at Cape Verde. Since we have classified the particle types in the companion paper, we can compare the $n_s$ during marine and dust periods. We added the following in page 22, line 3:

"CVAO is a place where marine and dust particles strongly intersect, and both particle types contribute to the surface area. In the companion paper, we have classified the aerosol at CVAO into four different types. Here, in addition to looking at average values as presented above, we selected the most clean marine (CVAO 1585) and most dusty (CVAO 1591) samples for a separate calculation of ns and added the results to Fig. 9. The $n_s$ is clearly higher for the sample collected during the dusty period than during the marine period at higher temperatures (roughly $>-16$ °C). However, at temperatures below $-18$ °C it is the other way around. In general, results for these vastly different cases are both still close to the upper limit of the parameterizations reported for SSA.

These comparisons to literature raise the question if and how $n_s$ should be used to parameterize atmospheric INP measurements, which, however, is a question far too prominent to be answered in this study. In general, it is still an open issue to which extent $N_{INP}$ can be parameterized, based on one or a few parameters, to reliably describe $N_{INP}$ for different locations around the globe. It might prove necessary to develop separate parameterizations for different locations or air masses, as it was already started for parameterizations based on particle number concentrations (see e.g., DeMott et al. (2010); Tobo et al. (2013); DeMott et al. (2015))."

22) Summary and Conclusions, page 21, lines 19-20: The sentence is not complete, and it is unclear if it is referring to other studies or this one. If referring to this study, I suggest to say that "biological particles appeared to contribute: : :" I note though that no confirmatory tests were performed to ascertain biological INP influence.

You were right about the confirmatory tests, which now were added, as said before: 
[revised manuscript text omitted]

**S2   Filter samples**

**S2.1   Background subtraction**

$N_{INP}$ from the field blanks was then subtracted from that of the filter samples, and the result was converted to background corrected atmospheric INP number concentrations, as the below equation shows:

5   $$N_{INP} = (-ln(1 - f_{ice,s}) + ln(1 - f_{ice,b}))/V \tag{S1}$$

The corrected atmospheric INP number concentration is $N_{INP}$, the frozen fractions measured for the filter samples and the field blanks are $f_{ice,s}$ and $f_{ice,b}$, respectively, and V is the volume of air sampled in each well.

**S2.2   CVAO PM$_{10}$**

**Table S2.** The information of PM$_{10}$ filter samples at CVAO, including sample number, start time, end time, duration, total sampling volume, sampling volume per well, sodium (Na$^+$) and chloride (Cl$^-$) mass concentration, total particle surface area concentration ($A_{total}$) and sample type.

| Sample Number | Start Time yyyy/mm/dd hh:mm:ss | End Time yyyy/mm/dd hh:mm:ss | Duration [minute] | Total Volume [std m$^3$] | Volume Per Well [std L$^{-1}$] | Na$^+$ $\mu$g m$^{-3}$ | Cl$^-$ $\mu$g m$^{-3}$ | $A_{total}$ $\mu$m$^2$ cm$^{-3}$ | Type |
|---|---|---|---|---|---|---|---|---|---|
| CVAO1583 | 2017/09/19 21:00:00 | 2017/09/20 21:00:00 | 1439.34 | 660.289 | 33.6882 | 4.40 | 6.19 | 370 | PM$_{10}$ |
| CVAO1585 | 2017/09/22 16:00:00 | 2017/09/23 16:00:00 | 1439.34 | 660.289 | 33.6882 | 3.09 | 4.97 | 89 | PM$_{10}$ |
| CVAO1586 | 2017/09/23 16:00:00 | 2017/09/24 16:00:00 | 1439.34 | 660.289 | 33.6882 | 2.36 | 3.36 | 78 | PM$_{10}$ |
| CVAO1587 | 2017/09/24 16:00:00 | 2017/09/25 16:00:00 | 1439.34 | 660.289 | 33.6882 | 2.83 | 3.54 | 158 | PM$_{10}$ |
| CVAO1588 | 2017/09/25 16:00:00 | 2017/09/26 16:00:00 | 1438.90 | 660.792 | 33.7139 | 3.32 | 4.98 | 277 | PM$_{10}$ |
| CVAO1589 | 2017/09/26 16:00:00 | 2017/09/27 16:00:00 | 1439.61 | 661.462 | 33.7481 | 1.41 | 1.99 | 159 | PM$_{10}$ |
| CVAO1590 | 2017/09/27 16:00:00 | 2017/09/28 16:00:00 | 1439.71 | 661.644 | 33.7573 | 1.77 | 2.70 | 198 | PM$_{10}$ |
| CVAO1591 | 2017/09/28 16:00:00 | 2017/09/29 16:00:00 | 1439.73 | 661.420 | 33.7459 | 5.04 | 8.41 | 325 | PM$_{10}$ |
| CVAO1592 | 2017/09/29 16:00:00 | 2017/09/30 16:00:00 | 1439.73 | 660.289 | 33.6882 | 6.49 | 11.26 | 297 | PM$_{10}$ |
| CVAO1593 | 2017/09/30 16:00:00 | 2017/10/01 16:00:00 | 1439.73 | 660.821 | 33.7153 | 5.32 | 8.99 | 238 | PM$_{10}$ |
| CVAO1594 | 2017/09/29 16:00:00 | 2017/09/30 16:00:00 | | | | | | | Blind filter |
| CVAO1595 | 2017/10/01 16:00:00 | 2017/10/02 16:00:00 | 1439.36 | 659.330 | 33.6393 | 4.52 | 6.67 | 172 | PM$_{10}$ |
| CVAO1596 | 2017/10/02 16:00:00 | 2017/10/03 16:00:00 | 1439.71 | 660.629 | 33.7056 | 3.71 | 6.49 | 171 | PM$_{10}$ |
| CVAO1597 | 2017/10/03 16:00:00 | 2017/10/04 16:00:00 | 1439.71 | 660.629 | 33.7056 | - | - | 169 | PM$_{10}$ |
| CVAO1598 | 2017/10/05 16:00:00 | 2017/10/06 16:00:00 | 1439.55 | 659.264 | 33.6359 | 2.58 | 3.33 | 162 | PM$_{10}$ |
| CVAO1641 | 2017/10/06 16:00:00 | 2017/10/07 16:00:00 | 1439.73 | 658.670 | 33.6056 | 4.67 | 6.91 | 244 | PM$_{10}$ |
| CVAO1642 | 2017/10/07 16:00:00 | 2017/10/08 16:00:00 | 1439.71 | 661.187 | 33.7341 | 5.46 | 8.54 | 271 | PM$_{10}$ |
| CVAO1643 | 2017/10/08 16:00:00 | 2017/10/09 16:00:00 | 1439.71 | 659.785 | 33.6625 | 5.22 | 7.98 | 230 | PM$_{10}$ |
| CVAO1644 | 2017/10/07 17:00:00 | 2017/10/08 17:00:00 | | | | | | | Blind filter |

[Figure]

**Figure S4.** $f_{ice}$ measured by INDA (without background subtraction) as a function of temperature in CVAO PM$_{10}$ filters. $f_{ice}$ of blind filters are shown by black dots.

[Figure]

**Figure S5.** $N_{INP}$ as a function of temperature from CVAO PM$_{10}$ filters. The field measurement of $N_{INP}$ in PM$_{10}$ by Welti et al. (2018) is shown by gray shadow. Error bars show the 95% confidence interval. Black dots show the measurement background.

[Figure]

**Figure S6.** Comparison of $N_{INP}$ as a function of temperature from CVAO 1596, CVAO 1641 and CVAO 1643 before and after heating (CVAO PM$_{10}$ filters). The field measurement of $N_{INP}$ in PM$_{10}$ by Welti et al. (2018) is shown by gray shadow. Error bars show the 95% confidence interval. Background correction is included for all filter samples.

[Figure]

**Figure S7.** Comparison of $f_{ice}$ measured by INDA (without background subtraction) as a function of temperature from CVAO 1596, CVAO 1641 and CVAO 1643 before and after heating (CVAO PM$_{10}$ filters).

**S2.3 CVAO PM$_1$**

[revised manuscript text omitted]

---

## Author Comment (AC2) · 4 Dec 2019

Dear Reviewer,

We thank you for doing this review and for your suggestions. Below, please find your original comments in blue and our responses in black. When referencing page and line numbers, we are always referring to the new versions of manuscript and SI, which are attached at the end. Concerning the literature cited in this answer to your review, we ask you to refer to the attached new version of the manuscript with tracked changes.

The manuscript presents an analysis of filter and water samples for their content of ice nucleating particles (INPs). Samples were collected at the Cabo Verde Island as part of the MarParCloud project.

What makes this study unique is that samples were simultaneously taken at sea and cloud level, and include filter samples of ambient air as well as cloud water and sea water samples. By comparing the INP content in these different compartments of the environment the authors infer the contribution of particles of certain origin and characteristics to the atmospheric INP population.

There are several points that need to be improved -- these are listed in detail below. Once these are addressed, the paper should be published in ACP.

Specific comments

1) P.1 ln.14f Attributing the difference in NINP from PM10 and PM1 samples to biological particles only based on the temperature of activation is speculative and over-reaching, eg. super-micron mineral dust particles can also cause ice formation at -10°C (see Hoose and Möhler, 2012). Are there indications against mineral dust particles being responsible for the higher activity? Were elevated NINP observed during dust events?

Thanks for your comments. We did additional measurements, as still filter material was left, and heated filter pieces from the three samples CVAO 1596, CVAO 1641 and CVAO 1643, i.e., those with high ice activity at high temperatures, at 95 °C for 1 hour. The elevated $N_{INP}$ at warm temperatures disappeared, as shown in the new Fig. 3. The respective values of $N_{INP}$ and $f_{ice}$ (only

for CVAO 1596, CVAO 1641 and CVAO 1643 before and after heating) are shown in the supplement, Fig. S6 and Fig. S7.

Page 1, lines 13-14 was changed to:

"After heating samples at 95 °C for 1 hour, the elevated $N_{INP}$ at the warm temperatures disappeared, indicating that these highly ice active INPs were most likely biological particles."

This supports the statement in the last sentence in this paragraph, which we hence did not change.

2) P.1 ln.18 Same as above. Most particles >1um are probably activated as CCN. There is no evidence for a biological origin of these INPs.

Based on the additional measurements mentioned at your comment above, we left this statement here as it was.

3) P.2 ln.7 In addition to homogeneous ice nucleation, heterogeneous ice nucleation by deposition nucleation is active also below -38°C. As the sentence is written it indicates that only homogeneous nucleation would be active below -38°C, which is incorrect.

Thanks for your comment. We changed it to:

"Ice crystals in the atmosphere can be formed either via homogeneous nucleation below −38 °C or via heterogeneous nucleation aided by aerosol particles known as ice nucleating particles (INPs) at any temperature below 0 °C."

4) P.2 ln.9 Are all droplets aqueous solutions?

Concerning droplets in the atmosphere, it can be assumed that all droplets are aqueous solutions.

5) P.2 ln.12 If there is a special aspect to the -20°C reported in Augustin-Bauditz et al., 2014 it should be mentioned. If not, the "below -15°C" from Hoose and Möhler, 2012 already includes -20°C and the Augustin-Bauditz et al., 2014 reference can be omitted. According to Fig. 3d in

Hoose and Möhler, 2012, super-micron mineral dust particles can be active INP already below -10°C and not -15°C.

Thanks for your comment. It was changed to:

"Submicron dust particles are recognized as effective INPs below $-20\,°C$ (Augustin-Bauditz et al., 2014) and super-micron dust particles were reported to be ice active even up to $-10\,°C$ (Hoose and Möhler, 2012; Murray et al., 2012)."

6) P.2 ln.17 The results in Boose et al., 2016 suggest desert dust to nucleate ice mostly below -25°C, especially airborne samples and the study shows differences among samples from different regions. This seems to contradict what is stated in ln.12-16. This section of the introduction should be revised, distinguishing mineral dust from desert dust studies and motivating the relevance of desert dust for the results presented here.

You are right, and we changed this part as follows:

"Boose et al. (2016) found that ice activity of desert dust particles at temperatures between $-35$ and $-28\,°C$ can be attributed to the sum of the feldspar and quartz content. A high clay content, in contrast, was associated with lower ice nucleation activity. In contrast to field measurements, in laboratory studies often separate types of mineral dusts are examined."

7) P.2 ln.26 Specify what is meant by "differences in desert sources".

Desert dust particles from different regions in a desert can have different mineral compositions. In order to specify, we changed this part as follows:

"Price et al. (2018) observed two orders of magnitude variability in $N_{INP}$ at any particular temperature from $\sim -13$ to $\sim -25\,°C$, which was related to the variability in atmospheric dust loading. This desert dust's ice nucleating activity was only weakly dependent on differences in desert sources, i.e., on the differences in mineral composition that particles emitted from different locations in the desert may have."

8) P.2 ln.28 Is NINP at ground level lower because the dust loading was lower at ground level?

Yes, it is. We added to the end of this sentence:"… level, where the dust loading was lower."

9) P.2 ln.32f This is incorrect. Bigg, 1973 suggested that INPs "are transported from a distant land source, or from a stratospheric source, and brought to sea level by convective mixing." Schnell and Vali, 1976 suggested a marine source could explain the observations of Bigg 1973.

Thanks for clarifying this. It was changed to:

"Based on a long-term measurement of INPs in the marine boundary layer in the south of and around Australia, Bigg (1973) suggested that INPs in ambient air were from a distant land source, or from a stratospheric source, and brought to sea level by convective mixing. Schnell and Vali. (1976) suggested a marine source could explain the observations of Bigg (1973)."

10) P.3 ln.2 What further evidence? This sentence seems to refer in a more general tone to the same study as the sentence before. Please clarify.

"Further evidence" was changed to "Furthermore".

11) P.3 ln.10f DeMott et al. 2010, 2015 suggest a correlation of NINP and the concentration of particles above a certain size. However they do not specify that only a fraction of large particles would act as INPs. Double check.

It is correct that these two publications rather describe parameterizations. However, as not all large particles act as INP (otherwise number concentrations of INP and all large particles would be the same), it is correct to say that only a fraction of these large particles is an INP.  We reworded this sentence as follows:

"Simultaneous measurements of $N_{INP}$ and particle number size distributions were used to develop parameterizations in which $N_{INP}$ depends on a temperature dependent fraction of all particles with sizes above 500 nm (DeMott et al., 2010, 2015)."

12) P.3 ln.20 The deduction that most biological INPs occur together with their original carrier appears incomplete. Clarify.

The explanation was meant to be given earlier in the sentence, which we now clarified as follows:

", but based on the fact that most atmospheric INPs seem to be super-micron in size, as observed in the above cited literature, it seems that most of the biological ice active macromolecules still occur together with their original carrier in the atmosphere."

13) P.4 first paragraph of Sec. 2.1.1. If the CVAO station is located on the northwest shore (ln. 11) of the island and wind direction is from the northeast (ln. 17), does this mean air first crossed the island before reaching the station? I think the CVAO is located on the northeast shore.

Thanks for catching this. CVAO is located at the northeastern shore of the São Vicente island. We changed this in the manuscript accordingly.

14) P.5 ln.16 It is mentioned in Sec. 2.1.1. that measurements took place during 31 days. Why where only 17 or 19 filters collected? How long was the sampling period per filter?

In the companion paper, the on-line measurement, including particle number size distribution, particle number concentration, CCN number concentration, started on 13 September and ended on 13 October. In order to keep consistency, we still used the same period in this paper. However, the filter samples were only collected from 19 September to 8 October. The sampling period was about 24 hours for each filter, with sampling times given in Table S1 in the supplement. We changed the beginning of Section 2.1.1. to:

 "The measurement campaign was carried out"

15) P.6 ln.29 By "brightness change" do you mean a change in transmitted light due to a change in opacity when droplets freeze?

You are right. The transmission light will decrease when the droplet freeze. As you can see in the image below, the dark circles mean the droplets were already frozen.

[Figure]

16) P.7 ln.8f Can you explain how the assumption of a Poisson distribution influences the result of Eq.1?

We are sorry but it is not totally clear to us what you mean, as Eq. 1 is derived based on the Poisson distribution as follows:

The Poisson distribution is a discrete probability distribution that expresses the probability of a given number of events occurring in a fixed interval of time or space if these events occur with a known constant rate and independently of the time since the last event (Haight, 1967). The probability of observing k events in an interval is given by the equation:

$$P_{(k)} = \frac{\lambda^{k} \cdot e^{-\lambda}}{k!}$$

Where

$\lambda$ is the average number of events per interval.

$e$ is the number 2.71828... (Euler's number) the base of the natural logarithms.

$k$ takes values 0, 1, 2, ....

$k! = k \times (k - 1) \times (k - 2) \times ... \times 2 \times 1$ is the factorial of $k$.

In the freezing array experiment, the probability of observing a certain number of INP in each droplet can be assumed to follow the Poisson distribution. And $\lambda = N_{INP}(\theta)*V$. Once there is at least one INP that is active at a certain temperature in a droplet, this droplet will freeze upon being cooled to this temperature, and then $P_{(0)} = 1 - f_{ice}(\theta)$.

In the Poison distribution:

$$P_{(0)} = e^{-\lambda}$$

Then we can get:

$$1 - f_{ice}(\theta) = e^{-N_{INP}(\theta)\cdot V}$$

$$N_{INP}(\theta) = \frac{-\ln(1 - f_{ice}(\theta)}{V}$$

17) P.7 sec.2.3.1. The principle of determining the cumulative INP concentration as well as the difference between INDA and LINA experiments are not very well explained. The explanation should mention that all wells contain multiple INPs but only the most active one (active at the highest temperature) causes freezing. The probability to find an active INP at a certain temperature increases with sample volume.

We added the following to page7, line 25:

"This way, the cumulative number of INP active at any temperature will be obtained although only the most ice active INP (nucleating ice at the highest temperature) present in each droplet / well will be observed."

Other than this, we already had said before that "The larger volume of water corresponds to a higher probability of the presence of INPs in each well, therefore INDA can detect INPs at warmer temperatures, … ." So nothing was changed in this regard.

18) P.7 Sec.2.3.2 Provide the equations you used for this calculation and some values to inform the reader how large statistical error and background are.

We included the equation and background results in the supplement page 6, lines 1-5.

19) P.7 ln.18 It is not explained previously that washing water was used or how it was prepared.

The "washing" is misleading here. We removed "washing".

20) P.7 ln.18 How was the information about the number of INPs per well obtained?

This is what is at the base of using the Poisson distribution, i.e., Eq. 1 will yield $N_{INP}$ based on the observed number of frozen droplets at each temperature where a measurement was made.

21) P.7 ln.20 A short explanation of the method from Agresti and Coull, 1998 would be helpful here.

We add the following in page 8, line 7:

"These confidence intervals were estimated according to the improved Wald interval which implicitly assumes a normal approximation for binomially distributed measurement errors."

22) P.8 ln.3f Can you provide a range of the derived freezing point depression for the samples. Was the freezing point depression experimentally confirmed, eg. by measuring the melting point depression?

The seawater salinity varied from 34.1 to 36.7 g $L^{-1}$. Based on this, according to theory, the freezing point depression varied from 2.1 to 2.3 °C.

We did not test the freezing point depression with seawater samples. Since the seawater samples are no longer available, we tested the freezing point depression of pure sodium chloride solution.

We dissolved 0.72 g sodium chloride in 20 mL MilliQ water to get a solution with a salinity similar to that of the SML and ULW samples. The frozen fraction ($f_{ice}$) of MilliQ water and of this sodium chloride solution are shown in the figure below. The error bars show 95% confidence intervals of $f_{ice}$. Due to large measurement uncertainties for the first frozen droplets, the freezing point depression should rather be determined from temperatures below approx. −25 °C, where, indeed, a freezing point depression temperature about ~2.2 °C was observed. It is therefore acceptable to use a freezing point depression of 2.2 °C in this study.

[Figure]

23) P.8 ln.9 The description of ns is imprecise. Ns gives the number of ice active sites per surface area, here the surface of all aerosol, ice active or not. Revise.

Following one of the first reviewer's (Paul DeMott) remarks, we extended this part to explain the $n_s$ for field measurement.

"For cases where a single type of aerosol, such as one type of mineral dust, is examined in laboratory studies, $A_{total}$ can be the total particle surface area. However, when field experiments are done, using the total particle surface area of the atmospheric aerosol assumes that all particles contribute to INP and have the same $n_s$, while the vast majority of these particles will not even be an INP. On the other hand, singling out the contribution of separate INP types in the atmospheric aerosol and relying $n_s$ only to them by using their contribution to the total surface area is at least demanding if not often impossible. This has to be kept in mind when interpreting heterogeneous ice nucleation in terms of $n_s$"

24) P.8 ln.20-26 This section is speculative and it is not clear why this comparison is relevant for the present study. Clarify.

So far, there are only a few publications in literature which discuss $N_{INP}$ in seawater. And while values for $N_{INP}$ in bulk water in these publications are similar to ours, those for SML are above ours, which, in itself, is an interesting observation. So we would want to keep this comparison here to show the $N_{INP}$ similarities and variations that can be there in seawater from different location of the world.

25) P.8 ln.27 Add an introducing sentence mentioning that the following analysis is done to compare NINP found in SML and ULW at Cabo Verde and explain why enrichment/depletion could be expected. Could INP in the SML originate from settling aerosol?

In page 9, lines 24-25 we already explained "To better quantify the enrichment or depletion of $N_{INP}$ in SML to ULW, we derived an enrichment factor (EF). The EF in SML was calculated by dividing $N_{INP}$ in SML ($N_{INP, SML}$) by the respective $N_{INP}$ measured in ULW ($N_{INP, ULW}$)." So it is not clear to us what more we could add here that has not already been said.

As organic material attaches to air bubbles rising to the surface (an effect well known in some industries and for aquaria, where it is used to clean liquids from organic contaminants), it may be expected that INP might also be enriched at the surface. And while settling of airborne aerosol

particles may contribute to INP in the SML, this contribution would only be very small, so that in general it will not play a big role. We added:

"An enrichment might be expected as organic material is known to attach to air bubbles rising to the ocean surface."

26) P.9 Fig.1 Why were samples 6, 7, 8, 10 and 11 not used? Instead of comparing to Wilson's data from a different environment, a direct comparison of SML to ULW by plotting both data on top of each other might show more clearly that NINP are the same between the two.

Samples 6, 7, 8, 10, 11 were collected for chemical measurements first. The remaining samples are not enough for INP measurement.

It is too crowded if we plot $N_{INP}$ of SML and ULW in the same figure.

27) P.9 ln.2 Clarify that you refer to the temperature at which NINP was determined in the drop freezing experiment. As is, it could be misunderstood as water temperature during sampling.

We changed this part as:

"Fig. 2 shows the EF as a function of the temperature at which $N_{INP}$ was determined in the freezing devices."

28) P.9 ln.3 Can you provide an explanation for the variation in EF with temperature? Does the interpretation change when considering the confidence interval of NINP? Do an error propagation of Eq. (5) and estimate the error in EF (should be included in Fig.2).

So far, it is still not clear which kind of particles in the seawater contribute INP. However, it can be assumed that there are several different ice active entities in seawater, which all can have different ice nucleation temperatures. So the variations in EF with temperature likely indicate that there are different amounts of these different types of ice active entities present in the different samples. Moreover, the many small wiggles in the curves displayed in this Figure are likely due to measurement uncertainties. To show this, and to follow your suggestion, we did the error

propagation for EF. This leads to a very busy plot, so we put this plot, i.e., the EF with error bars in the supplement, Fig. S3.

We added the following in page 9, lines 31-32:

"Most of the variation seen here is likely caused by measurement uncertainties, which are indicated in Fig. S3 in the supplement."

29) P.9 ln.8 Shouldn't SML thickness be related to concentration of dissolved organic matter? Explain why SML at Cabo Verde is larger than the SML in the Wilson et al., 2015 study even though conditions are oligotrophic.

The thickness of SML mainly depends on the collection techniques (Agogué et al., 2004;Aller et al., 2017). In Wilson et al. (2015), two techniques were deployed to collect SML samples. First, SML samples were collected into borosilicate glass bottles from a hydrophilic Teflon film on a rotating drum fitted to the 'Interface II' remote-controlled sampling catamaran, featured thickness around 20 μm. Secondly, SML samples were collected by glass plate, featured thickness around 80 μm. In this study, we collected the SML by using glass plate, with a thickness around 91 μm. Previous studies pointed out that rotating drum sampler and the glass dipping method probe different thicknesses of the SML, thus making a direct comparison of both SML thickness as well as enrichment factors generally difficult (Agogué et al., 2004;Aller et al., 2017). We added this information in page 10, lines 7-10.

30) P.10 Fig.2 add error estimation of EF.

See response to comment 28.

31) P.10 ln.9-11 What could be the reason for the variation between samples and why is the range of variation consistent to measurements at other locations?

$N_{INP}$ at one specific temperature is controlled by the number concentration of a mixture of different kinds of INP, which is explained in Welti et al. (2018). The fact that $N_{INP}$ at warm temperatures span 2 orders of magnitude is generally assumed to be due to a variation of the presence of

biological particles, which can serve as INPs. The commonality of this study and two cited experiments (Gong et al., 2019;O'Sullivan et al., 2018) is striking, and it may be assumed that land sources contribute to the INPs from long-range transport over broad regions, so that similar mixtures of INP can be present in different areas.

Could the number of samples or sampling duration determine the range of variation?

High variations in $N_{INP}$ have been observed from highly time resolved in-situ measurements (see e.g., Welti et al. (2018)), so that it can be assumed that longer sampling will somewhat smooth the detected concentrations. However, when comparing the ranges of values that are found with a number of different methods with different sampling times, high variations are found for all of the different methods (besides for Welti et al. (2018), also in DeMott et al. (2016) or in McCluskey et al. (2018b), which all include in-situ as well as filter based INP analysis). It can be assumed that there are times with either comparably persistently high or persistently low $N_{INP}$ on continental sites, in general. Even for samples that were collected for one or two weeks in the Arctic, a high variability in $N_{INP}$ was still observed, notwithstanding these long sampling times (Wex et al., 2019).

32) P.10 ln.12f Testing the heat sensitivity of the 3 samples could substantiate the interpretation that biological particles are responsible for the enhanced NINP.

Thank you for the suggestion. Samples CVAO 1596, CVAO 1641 and CVAO 1643 were heated to 95 °C for 1 hour and a large reduction of in $N_{INP}$ was observed. Based on this, we added one paragraph in page 12, line 3:

"Biological INPs contain specific ice-nucleating proteins. These proteins are disrupted and denatured by heating which causes them to lose their ice-nucleating ability. However, the inorganic ice-nucleating material, such as dust particles, is insensitive to heat (Wilson et al., 2015;O'Sullivan et al., 2018). Therefore, a commonly used heat treatment was deployed to assess the contribution of biological INPs to the total INPs in this study. Samples CVAO 1596, CVAO 1641 and CVAO 1643 were heated to 95 °C for 1 hour and the resulting NP are shown in Fig. S6. A clear comparison of before and after heating $f_{ice}$ is shown in Fig. S7. A large reduction of more than one order of magnitude in $N_{INP}$ at T> −15 °C was observed in the samples after heating. The reductions in $N_{INP}$ became smaller at colder temperature and were, for example, less than one order of magnitude at

T= −20 °C. This shows that biological aerosol contributed a large fraction of total INPs in PM$_{10}$ at T> −20 °C.”

33) P.10 ln.14ff. This paragraph is difficult to follow. Do you mean above -16.8°C concentrations within 2°C of each other are correlated? Looking at the data in Fig.3, NINP seem to change very little in 0.1°C steps, but are different between individual measurements.

In this paragraph, we describe the analysis of separate curves, i.e., we tested how well measurements from one sample were correlated with measurements of that same sample at different temperatures. We had already mentioned before:

““As long as the examined temperature difference was less than 2 °C, N$_{INP}$ were correlated.”

To make this clearer, the first sentence in this paragraph was changed to:

“The correlation of N$_{INP}$ at different temperatures within one sample was calculated, by comparing each INP at each temperature to that at each other temperature at which a measurement had been made. That was done separately for each of the samples.”

We added the tables that give all the values for $R^2$ and p for all temperatures for all different samples at the end of this response.

What is the actual regression model for which you report R2 and p value?

We used Pearson Correlation Coefficient.

What data was used for the regression?

The correlation of N$_{INP}$ at different temperatures was calculated.

Check the statistical power of correlating only few data points. Looking at the data in a differential spectrum (see Vali, 1971) might be a better method to identify temperatures where INPs of different origin become active.

The differential spectrum is good if only few different INP sources contribute (corresponding to clear bumps, i.e., a region of an increase in INP followed by a plateau), which is clearly explained in Welti et al. (2018). We have also used it for the analysis of INP from pollen in a laboratory study in the past (Augustin et al., 2013). In the present study, we only observed very few samples

(roughly 4 samples) with clear bumps at warm temperatures. Therefore, a differential spectrum is not a good way to generally characterize atmospheric INP sources, while it might work well close to strong INP sources or for laboratory studies.

34) P.11 ln.5 The dataset is by far not large enough to construct a robust pdf. The result in Fig.4 is vastly dependent on the choice of intervals to bin the data. The given pdf is therefore not suitable to perform any data analysis as the result could depend on the binning of data.

We deleted Fig. 4.

35) P.12 Fig.4 I suggest to remove Fig.4. It is not illustrating new information that is not already contained in Fig.3 and Fig.5. If you chose to keep Fig.4 check y-axis, the area under the pdf should be 1 or 100%.

We deleted Fig. 4.

PDF=$N_i$/$N_{total}$/$\Delta X$.

However, the total area was 1 for sure, but this could not easily be seen  as the bin width ($\Delta X$) was far smaller than 1.

36) P.12 ln.6ff Explain how the values in this paragraph were derived. I assume you compare the $N_{INP,PM10}$ to $N_{INP,PM1}$ from filters collected during the same time period and take the ratio?

Yes, that is what it is.

Page 12, line 31 was changed to:

"As for the first feature, we calculated the ratio of $N_{INP}$ in super-micron size range to $N_{INP}$ in $PM_{10}$ during the same time period and found that 83±22%, 67±18% and 77±14% (median±standard deviation) of INPs had a diameter of >1 μm at ice activation temperatures of −12, −15, and −18 °C, respectively."

37) P.13 ln.3ff Last sentence of this paragraph is speculative and repeating for the 6th time in this manuscript that high temperature activity of PM10 filter samples could be due to biological particles. As this seems to be a central point in your interpretation of the data I strongly recommend to experimentally test the heat sensitivity of NINP (eg. following the procedure described in Joly et al., 2014) to support that biological particles are causing the mentioned difference.

Done. See response 32.

38) P.14 ln.1 The difference in NINP (shown in Fig.6 (b)) is clearly visible above -20°C, not only above -17°C.

We said: "INPs that were ice active above ~ −17 °C were activated to cloud droplets to **a large degree**."

The difference in $N_{INP}$ becomes clearly visible above −20 °C.

Both of the statements are true and we would like to keep it as is.

39) P.14 ln.3f Is there evidence for a substantial fraction of droplets below 10um? Even though no direct observations are available from MV, observations in similar environments could help this discussion. Measurements of orographic cloud droplet distributions e.g. from Hawaii showed a bimodal droplet size spectra with both modes >10um (Squires, 1958).

We did not measure the cloud droplet size at MV. Indeed, according to laboratory (Chandrakar et al., 2016), model simulation (Igel and Heever, 2017) and field measurements (Miles et al., 2000;Siebert and Shaw, 2017), the cloud droplet size may smaller than 10 μm. Even in the Squires (1958), the paper you shared, Fig. 2 and Fig. 3 also indicated that some droplets have size is between 5 to 15 μm.

We cited one of the more newly published papers which talked about the droplet size distribution during the early stage of cumulus clouds, which is the closest to our conditions and added the following:

"These observations are consistent with results by Siebert and Shaw (2017) who observed broad cloud droplet size distributions in a size range from ~ 5 to 25 µm in shallow cumulus clouds, with the maximum of the distribution still being below 10 µm."

40) P.14 ln.5 This speculation is repeated several times throughout the paper but no evidence to support the biological nature of these INPs is presented. Either conduct heat sensitivity experiments and/or provide electron microscope images of large biological particles on the filters to demonstrate that this is a plausible interpretation, or delete the statement.

Done. Please see response 32.

41) P.15 ln.4ff In some cases over 100% difference in Na+ and Cl- concentration between the present study and Gioda et al., 2009 seem large and not comparable. In contrast to what other values do the authors think concentrations are comparable?

Thank you for the suggestion. It was changed to:

"Somewhat different values which are still roughly in the same range were reported by Gioda et al. (2009), who found in Puerto Rico the $Na^+$ and $Cl^-$ concentration in the cloud water varied from 3.79 to 15.53 and 5.90 to 23.20 mg $L^{-1}$, with a mean of 10.74 and 15.67 mg $L^{-1}$, respectively"

42) P.16 ln.13ff Couldn't Fcloud_air be estimated directly from the water collection rate of the CASCC2? This would reduce the uncertainty for the estimation of NINP.

The CASSC2 was sampling all the time while the cloud water sampling was intermittent. Therefore, it is not possible to calculate LWC from CASCC2.

43) P.17 ln.1f Does this range include the error estimation from the INP experiment? The two uncertainties (in Fcloud_air and NINP in cloud water) should be combined by error propagation when deriving the range of NINP,air.

This range did not include the error estimation from the INP experiment.

However, the INP experiment uncertainty can safely be assumed to be negligible.

Here is the reason:

Assuming a function contains a multiplication with two variables x and y.

$$f(x,y) = x \cdot y \qquad (1)$$

$$\delta f = \sqrt{\left(\frac{\partial f}{\partial x}\delta x\right)^2 + \left(\frac{\partial f}{\partial y}\delta y\right)^2} \qquad (2)$$

$$\frac{\partial f}{f} = \sqrt{\left(\frac{\delta x}{x}\right)^2 + \left(\frac{\delta y}{y}\right)^2} \qquad (3)$$

Now look at the function, $N_{INP,air} = F_{cloud\_air} * N_{INP,cloud}$. The uncertainties of $F_{cloud\_air}$ is at least 150% if we assume the median droplet diameter is 15 μm, with the variation from 7 to 20 μm. However, the uncertainties of $N_{INP,cloud}$ have a maximum of 80%, and go down to 40%. In function 3, if

$$\left(\frac{\delta x}{x}\right)^2 \gg \left(\frac{\delta y}{y}\right)^2$$

$$\frac{\partial f}{f} \approx \left(\frac{\delta x}{x}\right)$$

Which means the $N_{INP,cloud}$ uncertainties are negligible.

44) P.17 ln.3 The uncertainty range spreads over 2 orders of magnitude while the NINP cover 4 orders of magnitude. "general agreement" seems to have limited meaning here. The sensitivity on ddrop when calculating NINP,air from water samples determines the result.

As already suggested above, could the amount of collected cloud water be used to determine Fcloud_air or to constrain the ddrop range?

We agreed that the results from the calculations are very sensitive to the droplet diameter. However, the calculation based on the given size range of cloud droplet was chosen such that it covers all that can be expected to occur.

Alternatively LWC can be estimated from the CASCC2 collection rate (Sec. 2.3. in Demoz, 1996) for different drop size distributions (that could come from the literature eg. Sqires, 1958).

The collection rate from CASCC2 cannot be used to calculate LWC, as explained in question 42.

Another option to estimate LWC might be to use the NaCl content in cloud water and air as a tracer, similar to the method applied in Sec.3.4.

Thanks for your suggestion. We calculated the LWC by using the ratio of NaCl concentration in air to that in cloud water during the same period. We found this ratio varied from $1.1*10^{-7}$ to $4.2*10^{-7}$. The meaning of this ratio is the same as $F_{cloud\_air}$ in the paper, but based on different calculation methods. This ratio and $F_{cloud\_air}$ are comparable as they are in the same order of magnitude.

We added the following in page 17, line 23:

"To see how reliable these values are, we also examined the following: assuming all sodium chloride particles were activated to cloud droplets, $F_{cloud\_air}$ can be also estimated from the ratio of sodium chloride mass concentration in air to that in cloud water. This ratio varied from $1.1*10^{-7}$ to $4.2*10^{-7}$, which is at the lower end but still comparable to $F_{cloud\_air}$ as we derived it above."

45) P.17 ln.5 Instead of the collection efficiency at 3.5 um, it would be more useful to know the collection efficiency above the cut-off of the PM10 inlet of the filter sampler, where the droplet fraction not collected by the filter sampler but the cloud water collector should be found. According to Demoz, 1996 collection efficiency above 10um should be >80%. The high collection efficiency above 10um does not support the given explanation for a difference in NINP from filter and cloud water samples which is provided at the beginning of the paragraph (ln.3ff). Revise.

We are under the impression that you think that droplets were collected with through the $PM_{10}$ inlet onto filters, which is not the case. So that might already explain why you see an open issue here. The filter sampler providing the data shown in Fig. 7 was run on CVAO, so that potentially all INP were collected. On the other hand, the cloud water sampler, which ran at MV, does have a sampling efficiency below 100% through all sizes, starting with the mentioned 50% at 3.5 µm and, as you said, going up to above 80% at 10 µm. As we were at cloud base often, droplet sizes below 10 µm are to be expected, so that it can be assumed that not all droplets with INP were sampled. Particularly, if smaller droplets are collected with a lower collecting efficiency than larger droplets, the derived concentration will be lower compared to if all droplets were collected.

46) P.18 Eq. (7) and Fig.9 An error estimation for NINP from sea spray by error propagation of input variables in Eq. 7. Include error estimate in Fig.9.

Done.

47) P.18 ln.7 Did you use the individually measured NaCl concentration for each sampling period or the median to calculate the INP concentration in air? What is the range of the NaCl ratio on the right hand side of Eq.7?

The NaCl$_{mass,seawater}$ was very stable, with a median value ~31 g L$^{-1}$. NaCl$_{mass,air}$ showed large variability from 3.40 to 17.76 µg m$^{-3}$, with a median of 13.08 µg m$^{-3}$. We used the individually measured NaCl$_{mass,air}$ for each sampling period to divide the median NaCl$_{mass,seawater}$.

48) P.18 ln.10f Related to the previous comment. NINP at Cabo Verde and in the Arctic should be the same only if the NaCl ratio in Eq.7 is also the same. Alternatively, this highlights that the result of Eq.7 is largely insensitive to the NaCl ratio. This should be clarified.

Thanks for your comment. The NaCl ratios were both close to 10$^{-10}$ in this study and in the Arctic. Page 20, lines 2-3 was changed to:

"As discussed in section 3.1, N$_{INP}$ from ULW at Cape Verde are comparable to the Arctic, and the NaCl ratios were close to 10$^{-10}$ in both studies, therefore, $N_{INP}^{sea\ spray,air}$ (derived from ULW) are also comparable."

49) P.19 ln.1-12 It is unclear why enrichment of OC in SML is discussed here as no connection to NINP has been established. I recommend to delete this paragraph. All that can be said is that airborne NINP are higher than whatever NINP could have originated from the ocean.

It is correct that it is not clear to what extent INP are enriched in the SML. But it is very important to at least discuss a possible enrichment of N$_{INP}$ in SML to when information about airborne

concentrations are sought for. After all, we assumed the gap between sea spray $N_{INP}$ derived from SML in this study and McCluskey et al. (2018a) might be due to differences in the enrichment.

Since previous studies found that INPs in the ocean are associated with organic carbon (Wilson et al., 2015), here we used the enrichment of organic carbon in SML to air as reference. It is also clearly said that this is only an approximation in lack of better data: "It is not clear if INPs are included in the organic carbon for which the enrichment was observed." The discussion given at the location you refer to here gives the background for one of our main results and we would like to keep it as is.

50) P.20 Fig.10 I suggest to include ns for all temperatures covered by your experiments and for filter, water, CVOA, MV separately.

We did not have particle information in the water, so there is no way to calculated $n_s$ for water samples.

We did also not have an aerosol particle sizer (APS) at MV, which means we do not have information on super-micron particle number or surface area. As the main surface area is typically contributed from super-micron particles, we cannot derive $n_s$ at MV, either.

Also the box plot clearly shows the $n_s$ range, even although we only show $n_s$ at three temperatures. Adding $n_s$ for all temperatures will not change the results, and instead the new plot would look very crowded (we tried), so that we prefer to keep it as it is.

51) P.20 ln.9 Fig.10 should be motivated by stating what the expected ns are (SSA or dust) and then argue that the available ns parametrizations are not representative for Cabo Verde. It might be not surprising that ns parametrizations that are based on measurements in other environments do not capture the situation at Cabo Verde.

Thanks for your suggestion. We added the following in page 21, line 22:

"In the following, we will compare $n_s$ derived from our data with that from literature."

We added the following in page 21, line 32:

"These available $n_s$ parameterizations from previous literature may not be representative for Cape Verde, but we will still compare with them here."

Additionally, specify which data of "our data" is shown in Fig.10. Is it filter, water, CVAO or MV? As suggested in the previous comment all of these datasets could be of interest.

We changed this sentence:

"In Fig. 9, we show the surface site density derived for $N_{INP}$ from CVAO $PM_{10}$ filters (as shown by black boxes) following…"

52) P.21 ln.5f Here, the authors could suggest future directions, eg. regional, seasonal ns parametrizations or parameterizing NINP directly from field observations without employing surface area specific activity.

We added the following in page 22, lines 10-14:

"These comparisons to literature raise the question if and how ns should be used to parameterize atmospheric INP measurements, which, however, is a question far too prominent to be answered in this study. In general, it is still an open issue to which extent $N_{INP}$ can be parameterized, based on one or a few parameters, to reliably describe $N_{INP}$ for different locations around the globe. It might prove necessary to develop separate parameterizations for different locations or air masses, as it was already started for parameterizations based on particle number concentrations (see e.g., DeMott et al. (2010), DeMott et al. (2015) and Tobo et al. (2013))."

53) P.21 ln.19f Repetition of sentence from p.10 ln.12f. What could be the origin of these supermicron biological particles? Doesn't the evidence in this paper rather point to super-micron mineral dust?

Please see response 32.

54) P.21 ln.24 Provide evidence for biological particles or include dust as a possible source.

Please see response 32.

55) P.21 ln 26-27 Either, add figures showing both NCCN and PNSD at the two locations during cloud events and non-cloud events and provide difference in PNSD where the CCN active INPs are found, or, point the reader to Fig.8 in the companion paper. Was there a trend in NINP related to the particle types indicated in Fig.8 of the companion paper?

Thanks for your comment. We added "(see Fig. 8 in the companion paper)" in page 23, line 25.

Higher $N_{INP}$ generally appeared during dust periods and the lower $N_{INP}$ during marine type periods. High coarse mode particle number concentration is a sign for dust plumes. However, we did not find a good correlation between coarse mode particle number concentration and $N_{INP}$ and thus did not expand on this topic in the text.

Technical corrections

1) p.1 ln.4f SML and ULW might be sources of INPs, but sea and cloud level are compartments of the atmosphere where NINP are measured, not sources. Rephrase.

We replaced "sources" to "compartments".

2) P.1 ln.7 When mentioning "temperature" be specific in which system the temperature was measured. Here: "trends of EF with temperature." Temperature of what? Sea water, ambient or in the INP experiment?

It was changed to "with ice nucleation temperature" in page 1, line 7.

3) P.1 ln.8 Same as above. "at any particular temperature" could be understood as if sampling was conducted at different temperatures. The authors should be more specific and say: the temperature to which samples were exposed to in ice nucleation experiment.

We replaced "at any particular temperature" with "at any particular ice nucleation temperature".

4) P.2 ln.10 Freezing is not the same as ice nucleation. Immersion freezing refers to an ice nucleation mechanism rather than the freezing process. Rephrase.

Following Vali et al. (2015), the term "immersion freezing" can be used in this context: "Immersion freezing refers to ice nucleation initiated by an INP, or equivalent, located within the body of liquid." In the terminology by Vali et al. (2015), which is followed by many in the community, the one heterogeneous ice nucleation process that needs to be called "ice nucleation" is the deposition ice nucleation, as during that process no liquid water is required.

5) P.2 ln.15 Replace "more effective" with "more active" instead.

Done.

6) P.2 ln.22 Do you mean North African desert?

Yes. We changed "dust" to "desert".

7) P.2 ln.27 Replace "ice nucleating properties" by "NINP"

Done.

8) P.3 ln.1 Replace "INPs" with "the ice nucleation activity".

Done.

9) P.3. ln.22 Add: assuming that most INPs activate as CCN.

Done.

10) P.3 ln.23 Specify: "in rain samples"

This is cloud samples, not rain samples. We changed to "in cloud samples".

11) P.3 ln.31 Replace "for INPs analysis" with "to measure NINP"

We prefer to keep the formulation as it is, as we were not just measuring $N_{INP}$. We also characterized the INP contributions from the sub- and super-micron range separately and tried to link INP between sea- and cloud-water and in ambient air.

12) P.5 ln.4-6. Repetition of "specially designed". Delete in line 4-5.

Done.

13) P.5 ln.12 Is there something special about the Digitel filter sampler from the reseller Walter Riemer Messtechnik? If not the manufacturer should be referenced instead.

As far as we understood "Riemer Messtechnik" is the manufacturer (in the sense that the people there make the samplers we use), which is why it was mentioned here. The people from Riemer are the ones who come to us to introduce us to "how to use the instrument", and whom we contact for problems and repair issues, so Riemer is not a simple reseller.

14) P.5 ln.13 Move (Munktell, MK 360) to after "filters".

Done.

15) P.8 ln.12 Please add units to variables.

We apologize but it is not clear to us what you want, in this case. We only give a formula, here, with parameters, which are typically given without units (particularly as units might change, as e.g., the number concentration could be given in "per liter" or "per cubic meter" etc.).

16) P.8 ln15 Mention that this sentence refers to individual samples and starting from ln.17 variation between the 9 samples is discussed.

We added, at the end of the first sentence in this paragraph:

"… for both SML and ULW. Note that for each sample a separate INP spectrum is shown.

17) P.8 ln.26 Instead of "This" start the sentence with "The low biological activity in the SML around Cabo Verde"

Done.

18) P.9 Fig.1 Use the same y-axis scale for SML and ULW and include gridlines to facilitate comparing SML to ULW. Consider plotting the data on top of each other.

We changed the y-axis to the same scale. But plotting the data on top of each other is too crowd.

19) P.9 ln.7 Specify if you refer to sampling, or INP experiment technique.

We changed the sentence. Please see page 10, lines 5-10.

20) P.9 ln.10 Delete "the"

Done.

21) P.10 ln.7 "contribute" instead of "contributes"

We deleted Fig. 4. This sentence should also be removed.

22) P.10 ln.7 Replace "few" with "two"

Maybe you mean page 11, line7. It was changed.

23) P.13 Fig.5 Check unit of y-axis. Should not be %. Add to the figure caption what range is represented by the box and whisker of the boxplot. Due to the limited number of samples it would

be better to just provide the range instead of a boxplot (which requires the assumption of an underlying distribution).

We changed the y-axis label.

The boxes represent the interquartile range. Whiskers represent 10th to 90th percentile. We added this information in the figure caption.

24) P.13 ln.12, 14, 17 Avoid vague qualifiers "more or less", "only little", "quite similar", "mostly" and quantify instead.

Done.

25) P.13 ln.19 Add: "…obvious from Fig.6 that…"

Done.

26) P.14 Fig.6 Add (a) and (b) to the subfigures and add gridlines for easier readability. In the caption put the (a),(b) before describing the subfigure: "… MV PM10 filters during (a) less (cloud time fraction <10%) cloud effected periods and (b) highly…"

Done.

27) P.16 ln.25-28 Give the equation for this calculation.

Done.

28) P.17 Fig.8 First sentence in figure caption is incomplete. Replace "shown by" with "shown as".

Done.

29) P.18 Fig.9 Caption: "error bars showing" instead "error bars show"

The caption was changed to:

"Atmospheric $N_{INP}$ are shown as a function of temperature from $PM_{10}$ filters (black triangles), together with error bars showing the 95% confidence interval."

30) P.19 ln.26-28 Revise structure of sentence.

It was reformulated to:

"On the other hand, mineral dust is associated with a factor of 1000 higher ice surface site density (a measure to describe the ice activity per particle surface area), compared to SSA (Niemand et al., 2012; DeMott et al., 2016; McCluskey et al., 2018a)."

31) P.19 ln.31 Add: "… ice activity of super-micron mineral dust…"

This would not be correct, as also smaller dust particles can be ice active. Nothing changed.

32) P.20 ln.5 Add: "… associated with biological particles, but has also been observed for supermicron dust samples (Hoose and Möhler, 2012)."

Samples CVAO 1596, CVAO 1641 and CVAO 1643 were heated to 95 °C for 1 hour and a great reduction of in $N_{INP}$ was observed. We revised this whole paragraph, and the sentence you refer to here was deleted, so please refer to the new version of manuscript.

33) P.21 ln.4 You could add: "… do not originate from sea spray, but are dominated by supermicron dust."

We changed to: "do not originate from sea spray, but are dominated by super-micron dust and/or biological particles."

34) P.21 ln.9 Instead of "thorough analysis" specify what kind of analysis is shown in the companion paper.

As this is the summary of the present work and not the companion paper, this would make the summary unnecessarily longish. It is mentioned in more detail in the text above what was done in the companion paper, which should suffice, and we would therefore like to leave it as is.

35) P.21 ln.13f Freezing experiments with the devices used for this study should give reliable data up to 0°C and not "roughly" from below -5°C.

In this study, we got $N_{INP}$ from roughly −5 to −25 °C. So technically you are correct, but in fact it is only this reduced range for the present study, which we talk about in this sentence.

36) P.21 ln.15f Revise after correcting Sec. 3.1.

Done.

37) P.21 ln.21ff Revise after correcting Sec. 3.2.1.

We added the results of the heat treatment to the summary.

38) P.21 ln.25 "quite similar" should be put into perspective based on the limited number of investigated samples.

[revised manuscript text omitted]

**S2   Filter samples**

**S2.1   Background subtraction**

$N_{\mathrm{INP}}$ from the field blanks was then subtracted from that of the filter samples, and the result was converted to background corrected atmospheric INP number concentrations, as the below equation shows:

$$\quad N_{\mathrm{INP}} = (-ln(1 - f_{\mathrm{ice,s}}) + ln(1 - f_{\mathrm{ice,b}}))/V \tag{S1}$$

The corrected atmospheric INP number concentration is $N_{\mathrm{INP}}$, the frozen fractions measured for the filter samples and the field blanks are $f_{\mathrm{ice,s}}$ and $f_{\mathrm{ice,b}}$, respectively, and V is the volume of air sampled in each well.

**S2.2   CVAO PM$_{10}$**

**Table S2.** The information of PM$_{10}$ filter samples at CVAO, including sample number, start time, end time, duration, total sampling volume, sampling volume per well, sodium (Na$^+$) and chloride (Cl$^-$) mass concentration, total particle surface area concentration ($A_{\mathrm{total}}$) and sample type.

| Sample Number | Start Time yyyy/mm/dd hh:mm:ss | End Time yyyy/mm/dd hh:mm:ss | Duration [minute] | Total Volume [std m$^3$] | Volume Per Well [std L$^{-1}$] | Na$^+$ $\mu$g m$^{-3}$ | Cl$^-$ $\mu$g m$^{-3}$ | $A_{\mathrm{total}}$ $\mu$m$^2$ cm$^{-3}$ | Type |
|---|---|---|---|---|---|---|---|---|---|
| CVAO1583 | 2017/09/19 21:00:00 | 2017/09/20 21:00:00 | 1439.34 | 660.289 | 33.6882 | 4.40 | 6.19 | 370 | PM$_{10}$ |
| CVAO1585 | 2017/09/22 16:00:00 | 2017/09/23 16:00:00 | 1439.34 | 660.289 | 33.6882 | 3.09 | 4.97 | 89 | PM$_{10}$ |
| CVAO1586 | 2017/09/23 16:00:00 | 2017/09/24 16:00:00 | 1439.34 | 660.289 | 33.6882 | 2.36 | 3.36 | 78 | PM$_{10}$ |
| CVAO1587 | 2017/09/24 16:00:00 | 2017/09/25 16:00:00 | 1439.34 | 660.289 | 33.6882 | 2.83 | 3.54 | 158 | PM$_{10}$ |
| CVAO1588 | 2017/09/25 16:00:00 | 2017/09/26 16:00:00 | 1438.90 | 660.792 | 33.7139 | 3.32 | 4.98 | 277 | PM$_{10}$ |
| CVAO1589 | 2017/09/26 16:00:00 | 2017/09/27 16:00:00 | 1439.61 | 661.462 | 33.7481 | 1.41 | 1.99 | 159 | PM$_{10}$ |
| CVAO1590 | 2017/09/27 16:00:00 | 2017/09/28 16:00:00 | 1439.71 | 661.644 | 33.7573 | 1.77 | 2.70 | 198 | PM$_{10}$ |
| CVAO1591 | 2017/09/28 16:00:00 | 2017/09/29 16:00:00 | 1439.73 | 661.420 | 33.7459 | 5.04 | 8.41 | 325 | PM$_{10}$ |
| CVAO1592 | 2017/09/29 16:00:00 | 2017/09/30 16:00:00 | 1439.73 | 660.289 | 33.6882 | 6.49 | 11.26 | 297 | PM$_{10}$ |
| CVAO1593 | 2017/09/30 16:00:00 | 2017/10/01 16:00:00 | 1439.73 | 660.821 | 33.7153 | 5.32 | 8.99 | 238 | PM$_{10}$ |
| CVAO1594 | 2017/09/29 16:00:00 | 2017/09/30 16:00:00 | | | | | | | Blind filter |
| CVAO1595 | 2017/10/01 16:00:00 | 2017/10/02 16:00:00 | 1439.36 | 659.330 | 33.6393 | 4.52 | 6.67 | 172 | PM$_{10}$ |
| CVAO1596 | 2017/10/02 16:00:00 | 2017/10/03 16:00:00 | 1439.71 | 660.629 | 33.7056 | 3.71 | 6.49 | 171 | PM$_{10}$ |
| CVAO1597 | 2017/10/03 16:00:00 | 2017/10/04 16:00:00 | 1439.71 | 660.629 | 33.7056 | - | - | 169 | PM$_{10}$ |
| CVAO1598 | 2017/10/05 16:00:00 | 2017/10/06 16:00:00 | 1439.55 | 659.264 | 33.6359 | 2.58 | 3.33 | 162 | PM$_{10}$ |
| CVAO1641 | 2017/10/06 16:00:00 | 2017/10/07 16:00:00 | 1439.73 | 658.670 | 33.6056 | 4.67 | 6.91 | 244 | PM$_{10}$ |
| CVAO1642 | 2017/10/07 16:00:00 | 2017/10/08 16:00:00 | 1439.71 | 661.187 | 33.7341 | 5.46 | 8.54 | 271 | PM$_{10}$ |
| CVAO1643 | 2017/10/08 16:00:00 | 2017/10/09 16:00:00 | 1439.71 | 659.785 | 33.6625 | 5.22 | 7.98 | 230 | PM$_{10}$ |
| CVAO1644 | 2017/10/07 17:00:00 | 2017/10/08 17:00:00 | | | | | | | Blind filter |

[Figure]

**Figure S4.** $f_{ice}$ measured by INDA (without background subtraction) as a function of temperature in CVAO $PM_{10}$ filters. $f_{ice}$ of blind filters are shown by black dots.

[Figure]

**Figure S5.** $N_{INP}$ as a function of temperature from CVAO PM$_{10}$ filters. The field measurement of $N_{INP}$ in PM$_{10}$ by Welti et al. (2018) is shown by gray shadow. Error bars show the 95% confidence interval. Black dots show the measurement background.

[Figure]

**Figure S6.** Comparison of $N_{INP}$ as a function of temperature from CVAO 1596, CVAO 1641 and CVAO 1643 before and after heating (CVAO PM$_{10}$ filters). The field measurement of $N_{INP}$ in PM$_{10}$ by Welti et al. (2018) is shown by gray shadow. Error bars show the 95% confidence interval. Background correction is included for all filter samples.

[Figure]

**Figure S7.** Comparison of $f_{ice}$ measured by INDA (without background subtraction) as a function of temperature from CVAO 1596, CVAO 1641 and CVAO 1643 before and after heating (CVAO $PM_{10}$ filters).

**S2.3 CVAO PM$_1$**

[revised manuscript text omitted]

**S3 Cloud samples**

**Table S5.** The information of cloud water samples, including sample number, start time, end time, duration, volume, sodium ($Na^+$) and chloride ($Cl^-$) mass concentration and $N_{CCN,0.30\%}$.

| Sample Number | Start Time | End Time | Duration (h) | Volume | $Na^+$ | $Cl^-$ | $N_{CCN,0.30\%}$ |
|---|---|---|---|---|---|---|---|
| | yyyy/mm/dd hh:mm:ss | yyyy/mm/dd hh:mm:ss | [h] | [mL] | mg L$^{-1}$ | mg L$^{-1}$ | cm$^{-3}$ |
| Cloud01 | 2017/09/20 13:25:00 | 2017/09/20 18:20:00 | 4.92 | 185 | 8.44 | 15.51 | 551 |
| Cloud03 | 2017/09/26 19:00:00 | 2017/09/27 08:00:00 | 13.00 | 435 | 8.32 | 14.15 | 387 |
| Cloud04 | 2017/09/27 19:00:00 | 2017/09/28 07:30:00 | 12.50 | 544 | 5.00 | 9.27 | 239 |
| Cloud05 | 2017/09/28 19:00:00 | 2017/09/29 07:30:00 | 12.50 | 537 | 14.18 | 24.57 | 560 |
| Cloud11 | 2017/10/04 19:00:00 | 2017/10/05 07:30:00 | 12.50 | 150 | 46.11 | 70.30 | 481 |
| Cloud12 | 2017/10/05 07:45:00 | 2017/10/05 17:38:00 | 9.88 | 78 | 22.75 | 36.99 | 494 |
| Cloud13 | 2017/10/05 17:40:00 | 2017/10/05 20:10:00 | 2.50 | 133 | 16.97 | 25.23 | 442 |
| Cloud14 | 2017/10/05 20:10:00 | 2017/10/05 23:30:00 | 3.33 | 131 | 17.31 | 24.36 | 473 |
| Cloud15 | 2017/10/05 23:30:00 | 2017/10/06 04:00:00 | 4.50 | 120 | 21.85 | 31.95 | 491 |
| Cloud16 | 2017/10/06 04:05:00 | 2017/10/06 08:00:00 | 3.92 | 120 | 16.87 | 19.77 | 445 |
| Cloud19 | 2017/10/06 18:00:00 | 2017/10/07 06:30:00 | 12.50 | 537 | 18.34 | 29.10 | 482 |
| Cloud20 | 2017/10/07 06:48:00 | 2017/10/07 10:48:00 | 4.00 | 88 | 28.19 | 41.54 | 510 |
| Cloud24 | 2017/10/08 19:00:00 | 2017/10/09 07:00:00 | 12.00 | 537 | 24.54 | 32.46 | 625 |

[Figure]

**Figure S11.** Times during which MV was in clouds (in red shadows) and the sampling time of all cloud water and that of some selected CVAO PM$_{10}$ filters.

[Figure]

**Figure S12.** $f_{ice}$ measured by LINA as a function of temperature in cloud water.

**S4 Particle surface area size distribution**

A thorough aerosol characterization has been done during the measurement campaign, and is described in detail in Gong et al. (2019). Fig. S14 shows the median particle surface area size distribution (PASD) for the whole campaign. Error bars show the 75th and 25th percentiles. Two different modes were observed, i.e., a small mode (30-500 nm) and a larger mode (500 nm-10
5   $\mu$m). The larger mode particle surface area is about 3 times higher than the small mode. Based on the PASD, the concentrations for the total surface area of the particles were calculated. The total particle surface area concentration ($A_{total}$) varied from 35 to 824 $\mu$m$^2$ cm$^{-3}$, with a median of 116 $\mu$m$^2$ cm$^{-3}$. The averaged $A_{total}$ during each CVAO PM$_{10}$ sampling period varied from 78 to 370 $\mu$m$^2$ cm$^{-3}$ (summarized in Tab. S2). Based on airborne measurements in the Saharan dust layer, Price et al. (2018) found $A_{total}$ mainly above 100 with a maximum of 688 $\mu$m$^2$ cm$^{-3}$, which is higher than values found for this study, likely due
10   to the fact that Cape Verde is at some distance to the Sahara and also that less strong dust events were sampled.

[Figure]

**Figure S13.** $f_{ice}$ measured by INDA as a function of temperature in cloud water.

[Figure]

**Figure S14.** The median PASD during the whole campaign. The error bar indicates the range between the 75th and 25th percentiles.

---

## Referee Report (RR1)

**2$^{nd}$ Referee report on "Characterization of aerosol particles at Cape Verde close to sea and cloud level heights - Part 2: ice nucleating particles in air, cloud and seawater" by Xianda Gong et al.**

The authors have addressed most of the comments and improved the manuscript. However, I would like to suggest some further edits to improve the manuscript before it can continue the review process.

*General comments:*

The **abstract** is somewhat fragmented and could be constructed better to describe the work that has been conducted and highlight the outcomes. The section from line 8-17 on page 1 is mixing information of range, absolute INP concentration and potential nature of INPs. Also, in the last section page 2, line 1ff the first sentence seems disconnected from the rest.
Page 2, line 17, you mention a "feasible way to link $N_{INP}$ in air, ocean and cloud water". Do you mean here by determining and comparing NaCl masses in these compartments? If this is the case, please clarify this by adding a sentence.

**Active surface site density**. Could ns be estimated more accurate if you use the information that 80% of detected INPs are super-micron particles? If there are compelling reasons not to consider this information, it could be explained here.

**CVAO PM10**. Page 12, Line 13ff describes correlation of $N_{INP}$ at different temperatures. It seems trivial that a cumulative INP concentration is almost the same in a narrow temperature range for one sample. The authors need to explain and formulate more clearly how they reach the conclusion in the last sentence of this paragraph.

**3.3.2. Connecting INPs in the cloud water with these in the air.** The authors give information on the percentage of cloud time, sampling time and collected water volume. The flow rate through a CASCC2 can be found in Demoz et al., 1996 to be 5.8m$^3$/min. With this information it is straight forward to calculate the volume of cloud water per volume of air from the information given in Tab. S5:

$$F_{cloud\_air} = \frac{V_{water}}{sampling\ duration \cdot fraction\ in\ cloud \cdot flow\ rate}$$

Please confirm the results in this section by using this more direct estimation of $F_{cloud\_air}$.

*Specific Comments:*

Page 1, Line 3ff.: I suggest reformulating this sentence: "In this work, we examined $N_{INP}$ at Cape Verde in different environmental compartments: namely, the ocean sea surface microlayer, underlying water, cloud water and the atmosphere close to both sea and cloud level."

Page 5, Line 18: delete "was"

Page 8, Line 17: "First,..." is not followed by Second in the following. Delete.

Page 8, Line 27: Replace "This provides an opportunity" by "PNSDs were used"

Page 10, Line 8: delete "in" before "herein".

Page 10, Fig.2: I recommend to directly use Fig.S3 that includes the uncertainty estimation, instead of Fig.2.

Page 12, Line 22: replace "observation" by "inspection".

Page 12, Line 26: The data provides information on the abundance of INP at different temperatures and not INP efficiencies for individual particles. Replace "efficient" with "abundant".

Page 12, Line 31: replace "PM10" with "PM1".

Page 16, Fig.6 caption: Consider reformulating to: "$N_{INP}$ in cloud water as a function of temperature…Previous field measurements of $N_{INP}$ in cloud water by Joly et al. (2014) are shown as red box for comparison."

---

## Editor Decision (ED1)

Stockholm, 1 January 2020

Dear authors,

Thank you for your revised manuscript. The referees had another look at your revised work and added few more minor comments (see online comments from referee #1 and PDF from referee #2). In addition, I added a few more comments below after reading the manuscript once more.

Your manuscript will be ready for ACP after these minor comments have been thoroughly considered.

Thanks and kind regards

Paul.

Comments:

- Abstract: I agree with referee #2 that the abstract could be improved by writing it in a more compact way and highlighting the main results of this work.
- Page 3, line 19: Mention once more that you state median and standard deviation (usually people state the arithmetic mean together with the standard deviation, so this could potentially be confusing to the reader).
- Page 3, line 22: These sentence reads difficult. Maybe replace "feature" by "contain". In addition, add some more references if it is really "well understood".
- Page 4, line 10: Please add the ACPD reference for the Pinxteren et al. paper, which is already in ACPD.
- Page 7, line 11: Here and throughout the manuscript (e.g. page 10/line 4, page 14/line 14, page 16/line7, etc.): The beginning of the sentence should not start with abbreviations of section, equation, figure, etc. and the abbreviations should be harmonized within the text. Please have a look at: https://www.atmospheric-chemistry-and-physics.net/for_authors/manuscript_preparation.html
- Units: I'm a bit confused about the units. In Figure 1, you show '[# $L^{-1,water}$]' for the INP concentration within SML and UWL samples, while for the air samples you give '[# std $L^{-1,air}$]'. What does the 'std' stand for? Standard deviation? Standardized to STP? I guess later one since it is not shown for the water samples but then it should be behind the unit of liter of air. Please clarify this in the text somewhere. In addition, I would recommend to keep the mathematical power free of any add-ons like 'air' or 'water'. Maybe just move the 'air' and 'water' into the subscript and e.g. write '$L_{air}^{-1}$'.
- Page 17, line 10-16: Strictly spoken there should be units ($m^3_{water}/m^3_{air}$) for the volume of liquid cloud water per volume air.
- In general, I would say that the readability of some of the figures would be greatly improved if you would mention the full variable names to the figure captions.
- The conclusion could also be shortened to focus on the actual findings of this study (e.g. by removing the many references to the companion paper, which could be moved to the discussion section).
- The author contributions of Hartmut Hermann and Thomas Müller are missing.

Supplement:
- Figure S6: Aren't these data points already shown in Figure S5?
- Please also consider to improve the figure captions in the supplement by defining all the shown variable names and acronyms once more in the figure captions.

---

## Author Response (AR2)

Dear authors,

Thank you for your revised manuscript. The referees had another look at your revised work and added few more minor comments (see online comments from referee #1 and PDF from referee #2). In addition, I added a few more comments below after reading the manuscript once more.

Your manuscript will be ready for ACP after these minor comments have been thoroughly considered. Thanks and kind regards

Paul.

**Comments:**

- Abstract: I agree with referee #2 that the abstract could be improved by writing it in a more compact way and highlighting the main results of this work.
  This manuscript contains INP measurement for three environmental compartments, namely, the seawater, cloud water and atmosphere. For INP in atmosphere, we compared $N_{INP}$ close to sea and cloud level, and compared $N_{INP}$ in $PM_1$ and $PM_{10}$. Considering a great deal of useful information should be mentioned in the abstract, it is better to decompose lines 8-17 into short paragraphs to make important messages more clear to readers.
  Therefore, in the new version, paragraph 3 show $N_{INP}$ in $PM_{10}$ and heating treatment and paragraph 4 show the comparison of $N_{INP}$ in $PM_1$ and $PM_{10}$.

- Page 3, line 19: Mention once more that you state median and standard deviation (usually people state the arithmetic mean together with the standard deviation, so this could potentially be confusing to the reader).
  Done.

- Page 3, line 22: These sentence reads difficult. Maybe replace "feature" by "contain". In addition, add some more references if it is really "well understood".

We changed "feature" to "contain". Three additional citations were added in page 3, lines 26-29.

- Page 4, line 10: Please add the ACPD reference for the Pinxteren et al. paper, which is already in ACPD.

  Done.

- Page 7, line 11: Here and throughout the manuscript (e.g. page 10/line 4, page 14/line 14, page16/line7, etc.): The beginning of the sentence should not start with abbreviations of section, equation, figure, etc. and the abbreviations should be harmonized within the text. Please have a look at: https://www.atmospheric-chemistry-and-physics.net/for_authors/manuscript_preparation.html

  Done. Manuscript and supplement were checked to make sure that we use the right format.

- Units: I'm a bit confused about the units. In Figure 1, you show '[# $L^{-1,water}$]' for the INP concentration within SML and UWL samples, while for the air samples you give '[# std $L^{-1,air}$]'. What does the 'std' stand for? Standard deviation? Standardized to STP? I guess later one since it is not shown for the water samples but then it should be behind the unit of liter of air. Please clarify this in the text somewhere. In addition, I would recommend to keep the mathematical power free of any add-ons like 'air' or 'water'. Maybe just move the 'air' and 'water' into the subscript and e.g. write '$L_{air}^{-1}$'.

  "std" means the standardized STP. Since the STP is explained in Page 4, line 32, we changed the unit to "[# $L_{air}^{-1}$, STP]". In the manuscript, all "std" were removed. The "[# $L^{-1, water}$]" was changed to "[# $L_{water}^{-1}$]".

- Page 17, line 10-16: Strictly spoken there should be units ($m^3_{water}/m^3_{air}$) for the volume of liquid cloud water per volume air.

  We adjusted the units.

- In general, I would say that the readability of some of the figures would be greatly improved if you would mention the full variable names to the figure captions.

  Done.

- The conclusion could also be shortened to focus on the actual findings of this study (e.g. by removing the many references to the companion paper, which could be moved to the discussion section).

  We shortened the summary and conclusion section.

- The author contributions of Hartmut Hermann and Thomas Müller are missing.

  It has been mentioned that all co-authors proofread and commented the manuscript, which includes the two co-authors mentioned here.

Supplement:

- Figure S6: Aren't these data points already shown in Figure S5?

  We removed heated samples from Fig. S5. The purpose of Fig. S5 is to show the measurement background, as asked by reviewer #2.

- Please also consider to improve the figure captions in the supplement by defining all the shown variable names and acronyms once more in the figure captions.

  Done.

**Report #1:**

The authors have done an excellent job in responding to my review comments. Three tiny notes of no consequence.

1) Bigg (1973) mentions ocean sources at the top of the second column of page 1156 of his paper.

Thanks for your comment. This was corrected in the new version.

2) The freezing point depression demonstration is fine. I did not suggest it as a way of testing your experimental setup so much as because we have not found these exact values when diluting seawater samples (but always larger values by some amount).

Thanks for your comment, in any case. We had never tried this before and really liked that it fit so well.

3) Regarding the statement about parsing out the contributions of dust and marine source that, "using their contribution to the total surface area is at least demanding if not often impossible", it has at least been attempted under limited conditions (Cornwell et al., JGR, 124, 12,157–12,172, https://doi.org/10.1029/2019JD030466, 2019).

Thanks for your comment. Cornwell et al., 2019 is now cited in the manuscript.

**2nd Referee report on "Characterization of aerosol particles at Cape Verde close to sea and cloud level heights - Part 2: ice nucleating particles in air, cloud and seawater" by Xianda Gong et al.**

The authors have addressed most of the comments and improved the manuscript. However, I would like to suggest some further edits to improve the manuscript before it can continue the review process.

*General comments:*

The **abstract** is somewhat fragmented and could be constructed better to describe the work that has been conducted and highlight the outcomes. The section from line 8-17 on page 1 is mixing information of range, absolute INP concentration and potential nature of INPs. Also, in the last section page 2, line 1ff the first sentence seems disconnected from the rest.

This manuscript contains INP measurement for three environmental compartments, namely, the seawater, cloud water and atmosphere. For INP in atmosphere, we compared $N_{INP}$ close to sea and cloud level, and compared $N_{INP}$ in $PM_1$ and $PM_{10}$. Considering a great deal of useful

information should be mentioned in the abstract, it is better to decompose lines 8-17 into short paragraphs to make important messages more clear to readers.

Therefore, in the new version, paragraph 3 show $N_{INP}$ in $PM_{10}$ and heating treatment and paragraph 4 show the comparison of $N_{INP}$ in $PM_1$ and $PM_{10}$.

We deleted line 1 in page 2 in the new version.

Page 2, line 17, you mention a "feasible way to link $N_{INP}$ in air, ocean and cloud water". Do you mean here by determining and comparing NaCl masses in these compartments? If this is the case, please clarify this by adding a sentence.

Actually, with this statement we refer to Section 3.3.2, in which we mainly introduce a link using measured CCN number concentrations together with some assumptions. Then, in Section 3.3.2, we compare this link with the one obtained from NaCl and also from typical liquid water contents observed in atmospheric clouds (from literature). As these three approaches yield similar results, we then continue with our first approach. This is too much to be mentioned in the introduction, and we would prefer to not add anything so detailed so early in the text.

**Active surface site density**. Could ns be estimated more accurate if you use the information that 80% of detected INPs are super-micron particles? If there are compelling reasons not to consider this information, it could be explained here.

It would, in principal, be possible to separately estimate $n_s$ only for super-micron or only for submicron particles in this study. However, by doing this we will not get more accurate value for $n_s$ in general, so nothing would be gained, which is why we do not see an added value in reporting these results.

**CVAO PM10**. Page 12, Line 13ff describes correlation of $N_{INP}$ at different temperatures. It seems trivial that a cumulative INP concentration is almost the same in a narrow temperature range for one sample. The authors need to explain and formulate more clearly how they reach the conclusion in the last sentence of this paragraph.

You are totally right that a cumulative INP concentration is almost the same in a narrow temperature range for one sample. But we are rather aiming at temperature regions in which they do not correlate.

We said in the manuscript "In between these two temperature regimes (between >-16.8 °C and <-18.4 °C), the correlation of $N_{INP}$ was clearly lower. Therefore, it might be expected that INPs that are active in these two temperature regimes originated from different sources." For example, the $R^2$ of $N_{INP}$ at -13 and -18.5 °C is 0.37, which indicated that INPs that are active in these two temperature regimes originated from different sources.

**3.3.2. Connecting INPs in the cloud water with these in the air.** The authors give information on the percentage of cloud time, sampling time and collected water volume. The flow rate through a CASCC2 can be found in Demoz et al., 1996 to be 5.8m3/min. With this information it is straight forward to calculate the volume of cloud water per volume of air from the information given in Tab. S5:

$$F_{cloud\_air} = \frac{V_{water}}{sampling\ duration \cdot fraction\ in\ cloud \cdot flow\ rate}$$

Please confirm the results in this section by using this more direct estimation of $F_{cloud\_air}$.

You are right that this method could be used, too. However, using this method would (also) imply large uncertainties, originating first from the flow rate of the sampler (which was unfortunately, not well calibrated). Second, the exact fraction in cloud is difficult to come by, and as the mountain station often was close to cloud base, fluctuating in and out of the cloud, which introduces a large error. Third, the cloud droplet sampling efficiency is size dependent, from 20% at 2 µm to ~82% at 10 µm and above, and as no cloud droplet size distribution is available, a correction is not possible and a further large error would be introduced. Below we give the factor $F_{cloud\_air}$ derived with three different methods in a table. These methods are the one used by us in the manuscript, the method based on NaCl concentrations in air and cloud water samples and the method you introduce here. Using the method based on NaCl would yield INP concentrations that are ~ *0.23 of those we determined, using the method you suggest would yield INP concentrations that are ~ *2.4 larger than those we determined. Therefore, the

conclusions obtained from any of these methods will be the same, i.e., that there is roughly an agreement between number concentrations of airborne INP and INP in cloud water.

We therefore prefer to not add yet another method, particularly as this new one proposed here comes with large uncertainties.

| Cloud Sample | Based on $N_{CCN}$ | NaCl,air_median/NaCl,cloud | Based on CASCC2 flow |
|---|---|---|---|
| WW01 | 9.742E-07 | 4.423E-07 | 2.076E-06 |
| WW03 | 6.832E-07 | 1.512E-07 | 1.846E-06 |
| WW04 | 4.229E-07 | 3.137E-07 | 2.401E-06 |
| WW05 | 9.895E-07 | 3.473E-07 | 2.370E-06 |
| WW11 | 8.508E-07 | | 6.621E-07 |
| WW12 | 8.733E-07 | | 4.354E-07 |
| WW13 | 7.802E-07 | | 2.935E-06 |
| WW14 | 8.353E-07 | 1.419E-07 | 2.168E-06 |
| WW15 | 8.675E-07 | 1.099E-07 | 1.471E-06 |
| WW16 | 7.869E-07 | 1.614E-07 | 1.690E-06 |
| WW19 | 8.524E-07 | 2.443E-07 | 2.370E-06 |
| WW20 | 9.018E-07 | 1.662E-07 | 1.214E-06 |
| WW24 | 1.104E-06 | 2.317E-07 | 2.469E-06 |

*Specific Comments:*

Page 1, Line 3ff.: I suggest reformulating this sentence: "In this work, we examined $N_{INP}$ at Cape Verde in different environmental compartments: namely, the ocean sea surface microlayer, underlying water, cloud water and the atmosphere close to both sea and cloud level."

Done.

Page 5, Line 18: delete "was"

Done.

Page 8, Line 17: "First,..." is not followed by Second in the following. Delete.

Done.

Page 8, Line 27: Replace "This provides an opportunity" by "PNSDs were used"

Done.

Page 10, Line 8: delete "in" before "herein".

Done.

Page 10, Fig.2: I recommend to directly use Fig.S3 that includes the uncertainty estimation, instead of Fig.2.

The error bars that are shown in Fig. S3 cover some of the curves shown in Fig. 2 (less than half of all curves can be seen any more, in Fig. S3, and there is no way to change this as long as the uncertainty is included). But the main text rather discussed the separate curves than the measurement uncertainty. Therefore, we would prefer to keep it as it is now.

Page 12, Line 22: replace "observation" by "inspection".

Done.

Page 12, Line 26: The data provides information on the abundance of INP at different temperatures and not INP efficiencies for individual particles. Replace "efficient" with "abundant".

Of any number of super-micron particles, a much higher fraction will be an INP than of sub-micron particles. In that sense, "efficient" can be used here.

Page 12, Line 31: replace "PM10" with "PM1".

We checked the text carefully, but it is all correct in the original version.

[revised manuscript text omitted]
  (Leipzig Ice Nucleation Array) as a function of temperature in sea surface micro-layer (SML) and underlying water (ULW). All temperatures have been corrected for freezing point depression.

[Figure]

**Figure S2.** $f_{ice}$ measured by INDA (Ice Nucleation Droplet Array) as a function of temperature in SML and ULW. All temperatures have been corrected for freezing point depression.

[Figure]

**Figure S3.**  Enrichment factor (EF) as function of ice nucleation temperature. The EF=1 is shown by dashed line. Error bars show the measurement uncertainty.

**S2 Filter samples**

**S2.1 Background subtraction**

INP number concentration ($N_{INP}$) from the field blanks was then subtracted from that of the filter samples, and the result was converted to background corrected atmospheric INP number concentrations, as the below equation shows:

$$N_{INP} = (-ln(1 - f_{ice,s}) + ln(1 - f_{ice,b}))/V \qquad \text{(S1)}$$

The corrected atmospheric INP number concentration is $N_{INP}$, the frozen fractions measured for the filter samples and the field blanks are $f_{ice,s}$ and $f_{ice,b}$, respectively, and V is the volume of air sampled in each well.

**S2.2 CVAO PM$_{10}$**

**Table S2.** The information of PM$_{10}$ filter samples at  Cape Verde Atmospheric Observatory (CVAO), including sample number, start time, end time, duration, total sampling volume, sampling volume per well, sodium (Na$^+$) and chloride (Cl$^-$) mass concentration, total particle surface area concentration ($A_{total}$) and sample type.

| Sample Number | Start Time yyyy/mm/dd hh:mm:ss | End Time yyyy/mm/dd hh:mm:ss | Duration [minute] | Total Volume [std m$^3$] | Volume Per Well [std L] | Na$^+$ $\mu$g m$^{-3}$ | Cl$^-$ $\mu$g m$^{-3}$ | $A_{total}$ $\mu$m$^2$ cm$^{-3}$ | Type |
|---|---|---|---|---|---|---|---|---|---|
| CVAO1583 | 2017/09/19 21:00:00 | 2017/09/20 21:00:00 | 1439.34 | 660.289 | 33.6882 | 4.40 | 6.19 | 370 | PM$_{10}$ |
| CVAO1585 | 2017/09/22 16:00:00 | 2017/09/23 16:00:00 | 1439.34 | 660.289 | 33.6882 | 3.09 | 4.97 | 89 | PM$_{10}$ |
| CVAO1586 | 2017/09/23 16:00:00 | 2017/09/24 16:00:00 | 1439.34 | 660.289 | 33.6882 | 2.36 | 3.36 | 78 | PM$_{10}$ |
| CVAO1587 | 2017/09/24 16:00:00 | 2017/09/25 16:00:00 | 1439.34 | 660.289 | 33.6882 | 2.83 | 3.54 | 158 | PM$_{10}$ |
| CVAO1588 | 2017/09/25 16:00:00 | 2017/09/26 16:00:00 | 1438.90 | 660.792 | 33.7139 | 3.32 | 4.98 | 277 | PM$_{10}$ |
| CVAO1589 | 2017/09/26 16:00:00 | 2017/09/27 16:00:00 | 1439.61 | 661.462 | 33.7481 | 1.41 | 1.99 | 159 | PM$_{10}$ |
| CVAO1590 | 2017/09/27 16:00:00 | 2017/09/28 16:00:00 | 1439.71 | 661.644 | 33.7573 | 1.77 | 2.70 | 198 | PM$_{10}$ |
| CVAO1591 | 2017/09/28 16:00:00 | 2017/09/29 16:00:00 | 1439.73 | 661.420 | 33.7459 | 5.04 | 8.41 | 325 | PM$_{10}$ |
| CVAO1592 | 2017/09/29 16:00:00 | 2017/09/30 16:00:00 | 1439.73 | 660.289 | 33.6882 | 6.49 | 11.26 | 297 | PM$_{10}$ |
| CVAO1593 | 2017/09/30 16:00:00 | 2017/10/01 16:00:00 | 1439.73 | 660.821 | 33.7153 | 5.32 | 8.99 | 238 | PM$_{10}$ |
| CVAO1594 | 2017/09/29 16:00:00 | 2017/09/30 16:00:00 | | | | | | | Blind filter |
| CVAO1595 | 2017/10/01 16:00:00 | 2017/10/02 16:00:00 | 1439.36 | 659.330 | 33.6393 | 4.52 | 6.67 | 172 | PM$_{10}$ |
| CVAO1596 | 2017/10/02 16:00:00 | 2017/10/03 16:00:00 | 1439.71 | 660.629 | 33.7056 | 3.71 | 6.49 | 171 | PM$_{10}$ |
| CVAO1597 | 2017/10/03 16:00:00 | 2017/10/04 16:00:00 | 1439.71 | 660.629 | 33.7056 | - | - | 169 | PM$_{10}$ |
| CVAO1598 | 2017/10/05 16:00:00 | 2017/10/06 16:00:00 | 1439.55 | 659.264 | 33.6359 | 2.58 | 3.33 | 162 | PM$_{10}$ |
| CVAO1641 | 2017/10/06 16:00:00 | 2017/10/07 16:00:00 | 1439.73 | 658.670 | 33.6056 | 4.67 | 6.91 | 244 | PM$_{10}$ |
| CVAO1642 | 2017/10/07 16:00:00 | 2017/10/08 16:00:00 | 1439.71 | 661.187 | 33.7341 | 5.46 | 8.54 | 271 | PM$_{10}$ |
| CVAO1643 | 2017/10/08 16:00:00 | 2017/10/09 16:00:00 | 1439.71 | 659.785 | 33.6625 | 5.22 | 7.98 | 230 | PM$_{10}$ |
| CVAO1644 | 2017/10/07 17:00:00 | 2017/10/08 17:00:00 | | | | | | | Blind filter |

[Figure]

**Figure S4.** $f_{ice}$ measured by INDA (without background subtraction) as a function of temperature in CVAO $PM_{10}$ filters. $f_{ice}$ of blind filters are shown by black dots.

[Figure]

**Figure S5.** $N_{INP}$ as a function of temperature from CVAO $PM_{10}$ filters. Background correction of $N_{INP}$ is included for these filter samples. The field measurement of $N_{INP}$ in $PM_{10}$ by Welti et al. (2018) is shown by gray shadow. Error bars show the 95% confidence interval. Black dots show the measurement background.

[Figure]

**Figure S6.** Comparison of $N_{INP}$ as a function of temperature from CVAO 1596, CVAO 1641 and CVAO 1643 before and after heating (CVAO PM$_{10}$ filters). The field measurement of $N_{INP}$ in PM$_{10}$ by Welti et al. (2018) is shown by gray shadow. Error bars show the 95% confidence interval. Background correction of $N_{INP}$ is included for these filter samples.

[Figure]

**Figure S7.** Comparison of $f_{ice}$ measured by INDA (without background subtraction) as a function of temperature from CVAO 1596, CVAO 1641 and CVAO 1643 before and after heating (CVAO $PM_{10}$ filters).

**S2.3 CVAO PM$_1$**

[revised manuscript text omitted]